# Model for the Identification of Key Elements in the Management of Labor Relations and Conflict: Impact on the Internal Customer of Hotel Organizations and on Sustainable Development Goals 8

**María del Carmen Paradinas-Márquez** [1]**, José Antonio Vicente-Pascual** [2] **and Almudena Barrientos-Báez** [3,*]

1    Department of Business Management, ESIC University/ESIC Business & Marketing School,
     Av. de Valdenigrales, s/n, 28223 Madrid, Spain; carmen.paradinas@esic.university
2    Department of Market Research and Quantitative Methods, ESIC University/ESIC Business & Marketing
     School, Av. de Valdenigrales, s/n, 28223 Madrid, Spain; joseantonio.vicentepascual@esic.university
3    Department of Communication Theories and Analysis, Complutense University of Madrid, Av. Complutense, 3,
     28040 Madrid, Spain
*    Correspondence: almbarri@ucm.es; Tel.: +34-646491299

**Abstract:** The object of the study is to examine conflicts that occur in hotel companies, how they affect their structure, the quality of the work environment, the well-being of their workers, and their perception of the management of labor relations and conflict. For any company that wants to orient its efforts towards the fulfillment of the SDGs established in the 2030 Agenda, in this case, SDG 8 on decent work for all, it is key to understand which variables have the greatest influence on the management of labor relations and conflict. The aim is to identify those areas where they should focus their efforts to avoid organizational malaise that leads to economic and emotional costs derived from lower productivity and increased absenteeism, thus affecting their competitiveness. Information was collected to validate the objectives using a questionnaire with 57 items completed by workers with at least 5 years of experience in the sector. Using a simple linear regression, 10 key variables have been identified to explain the global satisfaction of employees, such as treatment, respect for company values, and the existence of defined processes and responsibilities.

**Keywords:** SDG 8; organizational management; conflict management; labor relations; tourism; hotel sector; absenteeism; psychosocial risks

## 1. Introduction

Human capital is an immaterial and intangible asset in hotel organizations, and since they are made up of people, according to Fried-Schnitman (2011) conflicts and crises are endemic to them. The important thing is how they are dealt with at the personnel and organizational management levels, some of these conflicts being acute and others chronic.

The objective of this article is to study conflicts in the workplace within tourism organizations, specifically those in the hotel sector, to understand the variables that affect it and to try to identify the importance of each of them in the general perception of conflict. It is reiterated that conflict is inherent and inevitable in human relationships in general, that every relationship has sufficient potential to cause conflict (Ramos Córdoba et al. 2013; Gómez Funes 2013; Carretero 2016; Arastey 2018). But, depending on the environment or the framework in which it occurs, it becomes necessary to analyze the origin of the conflict, the parties involved, the sector, and another series of factors to deal with it in the most appropriate way for its resolution within the multiple existing techniques (Constantino and Merchant 1997; Gómez Funes 2013).

In the same vein, the literature confirms that the very dynamics of group work lead to interactions between people in the workplace, and this leads to conflictive relationships

(Janssen 2004). But, depending on whether they are managed correctly, not managed, or badly managed, the relationship between the parties can be definitively destroyed or reinforced.

As pointed out in (Vinyamata 2004), there are multiple terms used to refer to conflict, its study, analysis, resolution, or management, such as mediation, conflict resolution, conflictology (understood as the science that studies conflict, coined in (Galtung 1995) as conflict management), conflict management, and alternative dispute resolution or ADRs (Alternative Dispute Resolutions).

This is why the concept of conflict "solution," which has always been understood as the elimination of conflict (either in an avoidance or win-lose aspect), is growing to incorporate its management into its scope, together with the identification of the interests of the parties and isolating them from their positions (Fajardo 2015). Companies linked to the tourism sector are facing new needs related to people management, so new formulas are emerging. New procedures are being created that lead organizations that use them to a privileged position in the management of change and give them an important competitive advantage.

This study contributes to both the academy and the business environment of the hotel sector as it relates the management of labor relations and conflict to the well-being of workers, establishing a link with SDG 8, focused on economic growth and decent work for all (United Nations 2023).

A poor perception by employees of their hotel's management of labor relations and conflict and a bad working environment can lead to sick leave caused by stress or depression, resulting in high levels of absenteeism that are unaffordable for the organization and that clash with SDG 8 in terms of decent work for all.

This paper contributes to identifying in a broad way those variables that have an influence on conflict management by organizations and, through an empirical study, allows the user to identify those that are critical in the perception of conflict and the management of labor relations by workers in the hotel sector, providing organizations with effective tools to avoid it or face it in a constructive way, and create positive environments.

The study begins with an extensive literature review on conflict, its elements, and the causes that trigger it, including the analysis of conflict in the workplace and its influence on organizations. Through this review, it is observed that organizations—in which conflict is not managed, is latent, or even explodes, triggering unwanted situations—face a bad working environment, to the point of causing sick leave and deeper problems in their workers. Hence, it is inferred that in companies where the management of labor relations and conflict is deficient, there are situations of absenteeism and turnover (Mohsin et al. 2023); this causes psychosocial risks, which means, in addition to a high cost for the organization, going against the achievement of the goals set in the United Nations Agenda 2030, specifically SDG 8: decent work and economic growth.

Subsequently, the definition of the seminal hypothesis of the work is included, centered on the existence of variables with a critical influence on the perception of labor relations and conflict management among all the elements analyzed. This is followed by a detailed explanation of the research process and the analysis of the results that allow the hypothesis to be contrasted. Finally, conclusions and limitations are drawn, and a proposal is made for the development of work through identifying future lines of research.

## 2. Conceptual Framework

### 2.1. Organizations Facing Conflict

The literature confirms that the very dynamics of working in a group cause interaction among people in the work environment, and this leads to conflicting relationships (Janssen 2004). But, depending on whether they are managed correctly, not managed at all, or managed badly, the relationship between the parties can be definitively destroyed or reinforced.

The analysis carried out here has focused not only on conflict resolution but also on conflict management, given that not all conflicts can be resolved, much less in a way that is

favorable for all parties involved. What happens then in these situations? Sometimes, conflicts arise within the organization, whether intergroup or intragroup (Benitez et al. 2012), which, although not resolved, can be managed so that they do not affect the productivity of employees, the team, and, in general, the parties involved and thus do not directly impact the company's costs or the health of workers.

The term conflict management is broader than conflict resolution as resolution expresses the reconciliation of interests, while conflict management includes both reconciliations and directing destructive conflicts towards constructive outcomes (Yirik et al. 2015).

As stated in (Burton 1998), conflict resolution can be considered the opposite of conflict management since the former seeks a permanent solution to the problem by accessing the sources, shedding light on the nature of the problem and its origin, thus eliminating the source of the dispute and preventing future confrontations.

Given that conflict has traditionally been perceived as a negative fact, which cannot be confronted because it could lead to more pernicious consequences than its avoidance, the need arises to contrast this with scientific-academic research, the approaches to conflict from its origins through its classification and its consequences (for the parties involved and for the environment).

Depending on the meaning we take of the concept of conflict, the way of dealing with it will vary substantially, and the way of dealing with these situations influences not only the possible benefits of the company or the working conditions but also the quality and health of the business structure itself (Merino 2008).

The negative meaning in the idea of conflict also has much to do with the confusion that usually occurs between the dispute and the means that are normally used to try to resolve it, such as shouting, fighting, or even the decision to go to court (Ramos and Hernán 2010), with the economic and emotional wear and tear that this entails. However, there are more and more recent theories that try to see the positive side of conflict, understanding it as something creative and capable of generating opportunity, growth, and development of new paradigms that push the participants not only towards personal growth but also to collaborate with social growth (Gómez Funes 2013).

Thanks to these studies, several ways of managing and preventing conflict are known that are used by individuals in their daily lives when faced with conflict, but, even in organizations, they are difficult to apply because, for most, it is a taboo subject that has to be avoided and even hidden, ignoring that it affects the well-being of the people who collaborate in them and, consequently, their productivity and corporate image. This especially affects companies that provide services, such as those that make up the hotel business fabric.

Any conflict can have negative or positive consequences; the crux is knowing how to respond with the appropriate means (Van de Vliet and De Dreu 1994; Rahim 2017).

As emphasizes Redorta (2018) in his constant research on conflict, it is difficult to enjoy precision in its study, either because what for one individual is a problem, for another is not, or because at the moment when the conflict arises, it is changing, it is active, it has its own evolution with multiple aspects that make its analysis complicated and not very rational due to the disparity of actors who have different degrees of intervention. And this is so because there are psychological and psychosocial components to the conflict that give it its dynamic character.

If we frame the positive view of conflict in the organizational environment (Janssen et al. 1996; Yeung et al. 2015), it will create positive results for the functioning of the company as the performance of its human capital will improve. Conflicts that bring benefit, and therefore functionality, tend to occur in the environment of dynamic companies that know how to adapt to change by maintaining creativity and making a positive analysis of the situation. On the contrary, in companies where there is a tendency to repress or excessively balance, it may become convenient to let the conflict emerge or even activate it (Van de Vliet and De Dreu 1994; Villamediana et al. 2015) since this would imply an opportunity to "loosen" the tendency to repress and would give those that are excessively

balanced the possibility of leaving their comfort zone and facing changes that, in both types of organizations, would give them the possibility of creating competitive advantages.

In Carnegie (2011), it is explained that, until not so long ago, conflicts that occurred in the workplace were considered simple disputes or more or less unpleasant mishaps of no great significance, but, little by little, they have been given the importance of their consequences, especially economic ones and those that affect the welfare of workers. The reality is that if they are not dealt with in a timely and appropriate manner, they will not only become entrenched but will also lead to a decrease in productivity, uneasiness, loss of direction, isolation of people (destroying the concept of group), and greater difficulty in coordinating them and, therefore, abandonment or sick leave due to psychosocial problems. On the other hand, if they are addressed in time and in good time, group cohesion, creativity, positive communication, exchange, and the feeling of belonging to the organization, which is so important for its proper functioning, will emerge.

In an international study in which one thousand five hundred company managers were surveyed, they were asked what skills they considered to be the main skills of a manager in the 21st century, with conflict prevention and resolution being the third most highly ranked (Urcola and Urcola 2019).

To detect the causes of the conflict, the person who is going to manage it can use, among other tools, questionnaires such as the one proposed in this study, which will be detailed when describing the methodology used. In fact, studies such as the one conducted (Mohammad et al. 2018) in on managers, assistant managers, and middle managers of tourism and hospitality companies conclude that they spend a lot of time on conflict management among their subordinates and that the practices put in place for this purpose are only slightly effective.

Redorta (2014) raise the question of the advantage of establishing behavioral patterns in the face of conflict rather than studying causality because each cause has its effect, and this can produce an infinite chain. However, the literature analyzing the causes of conflict is extensive, including the work environment studied here, and, therefore, reference is made to them.

Not only that hierarchization leads to conflict, but also specializations, from the moment that some employees may wonder why a certain sector of the organization is paid a bonus, why they have different working hours from the rest of their colleagues, etc., even though this is established in the Collective Bargaining Agreement. Comparative grievance is the cancer of labor relations.

When analyzing people's attitudes towards conflict, conflict situations are obviously not provoked to see how they react. Questionnaires and/or interviews are used to survey the way in which the group under study usually deals with conflicts or fictitious situations that are created (role play), but real conflict situations are not analyzed (Laca 2005).

In all countries, there are different ways of resolving conflicts, from the judicial route, arbitration, negotiation, conciliation, or mediation, all of them tending to resolve a dispute either directly by the parties involved (negotiation or conciliation) or through the intervention of a third party with greater or lesser involvement in the final decision (arbitration, conciliation, or mediation). Precisely one of the most propitious areas for the field of mediation in the labor field is the organizations, not only because there are conflicts within them that are not susceptible to go to the courts for their management, but precisely because early intervention can help not to judicialize many of the conflicts that occur within the company.

### 2.2. Management of Labor Relations and Conflict in Hotel Sector Companies

All stakeholders have expectations of the organization, including the internal customer, and when this externalizes the failure to meet their expectations, the reputation of the company can be significantly damaged to the point of compromising its viability (De Rueda 2018). In the hotel industry, it is common to focus resources on external customer satisfaction and pay little or no attention to the internal customer. Keeping the customer

happy and avoiding situations of conflict with them that could damage the reputation of the organization on the market is one of the main objectives of hotel companies.

Companies that focus on the internal customer, in addition to establishing conflict prevention and management mechanisms, try to train their employees in those skills, not only cognitive, that can improve productivity, but above all, the work environment and the feeling of belonging to the organization. The success of companies operating in the tourism sector is determined in part by the people who are part of them, so it is essential to know their aspirations, needs, and skills so that they can provide their best service in the most appropriate position (Baum 2007).

It is essential that organizations dedicated to the hotel sector have human capital that feels committed because the fact that the customer experience is positive is closely related to that commitment (Yohn 2016).

The fact that the experience regarding the user's interaction (Ruizalba Robledo et al. 2015) with the people who provide services at the hotel is negative may be due to many factors, among them the existence of a latent internal conflict in the organization that is not managed or that has exploded and has not been resolved. In fact, given the intangible nature of service provision, quality is sometimes measured mainly by the customers' opinion of that service (López Fernández and Serrano Bedia 2001). If the organization makes every effort to avoid conflicts with the external customer and ensures that, in case there are any, they do not escalate, even if it costs money, does it not seem to be common sense to exert the same effort with the internal customer? As a result of all that has been seen, there is a clear interest on the part of companies toward their personnel, implementing policies aimed at internal customer satisfaction, but although it seems obvious, there is no real effort to implement systems to manage and resolve conflicts, which also clashes with the achievement of decent work that does not affect the health of workers, as pointed out in SDG 8.

If the workers at a hotel are in a tense atmosphere, do not feel listened to in a conflict situation, and the environment is unfavorable to the perception of conflict as an opportunity, there will be an unfavorable working environment. This will trigger a series of detrimental events for the company, which will affect the external customer and be reflected in the costs; therefore, it is necessary to act on the improvement of relations at work.

The hypothesis of this paper arises from the theoretical context reviewed, from which it is clear that in organizations, there are a number of factors that directly influence labor relations and conflict, and that if they are well valued by the people working in the company, it will lead to a competitive advantage. In addition, how people hired by the company perceive certain variables will provide an overall view of how they feel about their work, how much they are willing to give of themselves, and the degree of attention that, in this case, the hotel sector, is paid to human capital.

The hotel industry is a sector with very particular characteristics in terms of employability, and, in addition, it has a great weight in the GDP of the Spanish economy. For this reason, analyzing the state of labor relations and conflict management in the hotel sector and the perception of its human capital is both a challenge and a motivation.

Many sectors have been affected by the coronavirus, but the hotel sector, along with the rest of the tourism-related activities, was hit hard by a situation unknown until now (UNWTO 2020).

### 2.3. The Cost of Conflict to the Organization

It is common for companies to consider the introduction of conflict management techniques, such as mediation or the change in organizational culture to introduce values such as the Culture of Peace, etc., as an increase in costs without taking into account the return on investment, or ROI (return on investment), from which they can benefit, since the benefit obtained with that investment, or the profit obtained, will be higher than the investment made.

This is because the cost of conflict within the organization can be much higher than the investment in conflict management and resolution tools due to conflict in the courts or latent and unresolved or poorly resolved disputes, leading to possible complaints to the labor inspectorate, sick leave, lower staff productivity, etc., which increases the company's costs (Ramírez 2013).

There are a series of causes that imply an increase in costs for the company if the conflict is not managed or is managed in an inadequate manner; in the event of an escalation of the conflict, it is very likely that it will end up in court, so it will be necessary to incur in counseling expenses, travel to conciliation acts, trial sessions, and possibly the imposition of legal costs, in addition to the cost of absenteeism of all those people who work for the organization and who must be absent to attend the trial. In fact, in the few cases in which the cost of the conflict for the company is studied, it is conducted based on the cost of resolving it only in court.

On the other hand, when personnel are dissatisfied, this dissatisfaction can lead to states of anxiety that result in sick leave. This means that in order to cover these absences, new personnel will have to be hired through a selection process that can sometimes be very costly, and once the person has been selected, a period of training and adaptation will have to be established, which also entails an economic cost.

Nor can one ignore the fact that when there is conflict in the work environment between individuals or between teams, time must be spent managing it, usually by superiors or the personnel in charge of the team; time that will be taken away from doing the work for which they have been hired and for which they are paid, time that will be referred to later.

In addition, the people around the dispute, even if they are not directly involved in it, will suffer distractions that will result in lost time and decreased productivity.

According to LinkedIn Talent Solutions (2019), and its published study on resources to improve employee retention, the estimated cost of replacing someone who has left the company ranges from 50% to 200% of their annual salary.

As stated by Sanz and Miralles (2017), people employed in U.S. companies spend, on average, two hours and thirty-eight minutes a week handling conflicts originating at work. This implies an approximate cost of three hundred and fifty-nine billion euros, calculated for the working population and based on the average price at which an hour of work is paid.

Previously, Femenia (2006) provided interesting data on the cost of conflict in the organization, such as, for example, that the top and middle management of organizations (directors and managers) spend between 30% and 50% of their working day dealing with conflicts.

This cost should not only be considered in terms of working hours (price/hour) and therefore as a loss for the organization, but it also means an economic cost for the people who work in it since many of them receive part of their salary in bonuses. These depend on their performance and objectives, so they will hardly be paid if they have to spend so much time dealing with conflicts instead of doing their work.

Returning to the cost for the organization, it is also important to pay attention to the implication for the organization of the loss of quality in decision-making since, if the conflict is latent, communication and information suffer biases that affect decision-making, and an organization that is governed by biased and hasty decisions will suffer, in the long run, loss of competitive advantage that will lead to an increase in economic costs.

Another consequence of hasty and defective decision-making is the loss of talent, either through dismissal, since it will be necessary to look for those responsible, or through voluntary resignation from the company by those who face the stress of having to make decisions under these conditions (Novel 2012), with all that this implies, as previously analyzed.

From what has been pointed out so far, it can be deduced that conflict in organizations leads to a high cost and that these costs can be classified into direct and indirect costs, as explained in the following figure.

According to Die (2013), direct costs are understood as the direct allocation of resources that must be dedicated to conflict management, and indirect or derived costs are understood as those arising from conflict and its management (the worse the conflict, the more costs it incurs).

Up to this point, the economic cost of conflict in organizations has been discussed, but the emotional cost of conflict cannot be ignored.

Organizations dedicated to the provision of services, such as those in the hotel sector, in their eagerness to perfect the interactions that take place between their staff and customers, neglect the interest in knowing how the employees who work there feel.

To define this need for self-control by the organization's personnel, another term called "emotional labor" has been articulated, which has come to demonstrate that, during service delivery, employees use a significant number of strategies aimed at regulating their emotions (Cho et al. 2013) in order to achieve compliance with the standards required by the company (Daus and Ashkanasy 2005).

Thus, emotional labor is described as having the "cost" and the requirement that the company's collaborators show positive emotions. That cost can become a drain on resources, hinder the completion of tasks, and threaten the well-being that should prevail in the work environment (Grandey et al. 2015); thus, hindering the achievement of SDG 8 in the long term.

From the moment a person is hired by a company, not only does an employment relationship begin in which the individual performs a job, but also begins an interaction with the rest of the people who make up the organization, and even, most likely, with third parties indirectly related to it. This interaction goes beyond the mere contractual sphere, bringing emotions, affections, and other psychological aspects into play.

This is because the relationships generated in the work environment have a considerable influence on people's lives and can affect them both positively (if they are good) and negatively (if they are bad). It is, therefore, necessary that the "win-win" objective comes into play in these relationships, which were created in the company environment, in order to create a system of dialogue that facilitates these relationships (Planès 2014).

### 2.3.1. Relationship between Conflict and Productivity

As we have seen, there are several factors that affect the additional costs incurred by the organization as a result of the conflict, and one of them is productivity, or more accurately expressed, the decrease in productivity that the conflict causes in the company. For this reason, this section will be brief, but it has been considered that, given its importance for the organization, a special section should be devoted to it in order to provide relevant data for this study.

When the quality of work life is good, it has been shown that the quality of the product or service is good (Elizur and Shye 1990). So, in the case of a hotel company, if the person who provides the service and is facing the public considers that his work environment is good, he will be productive and will turn to the satisfaction of the public.

As Vinyamata (2002) argue, billions of euros are lost as a result of the decrease in productivity directly related to the time spent in conflict.

Thus, if the environment is negative, and if the behavior in the face of difficulties is not positive, people will become demotivated, and productivity will decrease, as reflected in the analysis of the economic impact of conflict in the company.

This is evidenced by authors such as Ortiz-Campillo et al. (2019) when concluding that a good work climate has positive consequences such as increased labor productivity, low turnover, satisfaction, etc., and that, on the contrary, a bad work climate causes negative effects such as low productivity, maladjustment, and even losses of a socioeconomic nature for the company.

In the same sense, Folberg and Taylor (1992) state in their work that "unresolved conflicts between workers or between workers and management result in a loss of productivity."

According to a study by Harvard University's Conflict Management Center (Vinyamata 2005) productivity is reduced by 20% as a result of conflicts within the organization.

On the other hand, the Evaluation Report on the Measures for the Rationalization and Improvement of the Management of Temporary Disability, carried out by the State Agency for the Evaluation of Public Policies and the Quality of Public Services in 2009, shows that absence from work as a result of Temporary Disability due to Common Contingencies (ITCC) causes a decrease in productivity in companies.

The estimated cost, in 2005, of the decrease in productivity due to this type of sick leave, which will be dealt with in depth later on, is between 10,200 and 11,108 million euros, an amount that is difficult to assume for a country like Spain, where the vast majority of the business fabric is made up of small and medium-sized companies.

Interestingly, 14.3% of the people surveyed, when explaining the main reasons for requesting this leave, not being unable to work or suffering from any illness, claimed that it was due to situations of stress, depression, or anxiety; 4.5% to the bad working environment and 3.9% to situations of conflict in their work environment.

### 2.3.2. Labor Risks Associated with Labor Relations and Conflict Management

Every organization in Spain is obliged to maintain an action protocol on occupational risk prevention according to Law 31/1995, of 8 November 1995 on Occupational Risk Prevention, which implies the need for the employer to respond to any risk situation that arises within the organization, including difficulties in relations between its staff, all in clear accordance with the realization of decent work as set out in SDG 8.

As far as this thesis is concerned, the relevance of organizational conflict management, from the point of view of occupational risk prevention, will focus on the prevention of psychosocial risks, given that the damage caused in this area of risk prevention is due, among other factors, to the lack of proper management of the relationships that occur in the work environment.

Therefore, the first thing that is interesting to note is the fact that neither in the European nor in the national framework are psychosocial risks specifically contemplated, normatively speaking, as a matter of prevention. Only in the Royal Decree 39/1997 of January 17, approving the Regulation of Prevention Services, the psychosocial aspect is contemplated, but in relation to the training required for Senior Prevention Technicians, who must have as one of the required specialties, there is training in psychosocial risks, which is probably the first reference to be made legally in this regard.

On the other hand, since 1984, the International Labor Organization (ILO) has been stressing the importance of considering these psychosocial factors at work because of their influence on the physical and mental well-being of workers.

The ILO understands that "psychosocial stressors" include not only certain physical aspects but also those related to "work organization and systems, as well as the quality of human relations in the enterprise. All these factors interact and have an impact on the psychosocial climate of the enterprise and on the physical and mental health of the workers".

In this report, the ILO already highlights that certain conflicts, such as conflicts of authority or conflicts of competence, are psychosocial factors that have a negative influence on the health of workers. Therefore, it is clear this can be extrapolated to any type of conflict in the company.

Moreno (2011, p. 4) makes a distinction between psychosocial factors, which could include "corporate culture, work climate, leadership style or job design" whose impact can be positive or negative, and psychosocial risk factors of negative influence, which would refer to the conditions of the organization and trigger job stress and tension (Benavides et al. 2002). In this regard, it is interesting that the reference made in (Arastey 2018) to the judgment of the French Supreme Court, No. 11—18208, of 17 October 2012, argues mediation as a measure of prevention of psychosocial risks insofar as the company must protect not only the physical health but also the mental health of its employees, by

adopting the relevant protective measures. The High Court held the defendant company liable for having allowed a conflict situation to develop between two of its employees until it escalated into harassment and for not having adopted any measure to resolve it, while understanding that it was obliged to do so.

Authors such as Leka and Jain (2010), together with the World Health Organization (WHO) emphasize the need to establish early detection systems for those factors that may pose psychosocial risks.

Therefore, if conflict in organizations is a psychosocial risk factor, resolution and management systems should be established, especially aimed at prevention, so that it does not become an effective psychosocial risk that seriously affects the health of the people in the organization, contrary to SDG 8.

For example, during service, in organizations such as hotels, workers must constantly control their emotions and not allow them to be seen by customers (Ekman 1973). These rules imposed by the organization increase the pressure to provide this type of service (Rohrmann et al. 2011); hence, there are authors who state that interacting with the customer is psychologically exhausting (Walsh and Bartikowski 2013).

And when a person faces a psychologically exhausting day-to-day workday due to the need to control his emotions constantly in front of the customer, it will be more difficult for him to do so in front of his colleagues, so disputes will arise between them more easily than in other types of work. Therefore, the promotion of measures to mitigate these conflict situations by the organization will have an impact, among other things, on the health of its staff.

For Hochschild (1983), a pioneer in specifically dealing with this issue, the problem stems from the inevitable connection between emotions and their reflection in body expression and gestures that provoke those emotions.

The client not only asks for a perfectly prepared cocktail but also wants it to be served in a way that produces a unique experience that even shows that the person is enjoying serving him, which is a requirement of the organization placed on the person who is providing the service. Its corporate image is at stake (Martínez-Iñigo 2001), so hotel organizations not only look for experience or good appearances in the people they hire but also demand a certain attitude (Moreno 2011), although sometimes the price paid by employees is not only physical damage but also mental (Richards and Gross 1999; Wong et al. 2006). Terms surpassing even stress have been coined, such as burnout syndrome, first coined by Freudenberger and Richelson (1985).

Recognized as a disorder caused by chronic work-related stress, the World Health Organization (WHO) has introduced it into the list of diseases (International Statistical Classification of Diseases and Related Health Problems (ICD-11)). A worker may be diagnosed with burnout syndrome and, for this reason, may be dismissed from work since suffering from this disease or disorder causes demotivation and a distant attitude (depersonalized) towards work, emotional exhaustion, perception of not performing tasks correctly, rather ineffectively, and loss of other skills such as communication, etc.

The proof that this syndrome especially affects people who provide services, whose task consists of working with people (social workers, health workers, or educators), is that it was originally diagnosed for this type of people, although it has been extended to other professions (Salanova et al. 2000).

Finally, it is worth mentioning the European Working Conditions Survey 2020, conducted among 45,000 people in 36 countries by the European Foundation for the Improvement of Living and Working Conditions, Eurofound. It shows a tendency to examine more and more factors related to psychosocial aspects and some others related to those discussed here that have to do with social, organizational, and personal conflicts. Thus, for example, questions from the previous edition are maintained, such as "Does your job require you to hide your feelings?" or "Does your job involve dealing with angry customers?" or "Does your job involve handling emotionally disturbing situations for you?" (Parent-Thirion et al. 2015).

This shows a growing concern for the well-being of workers coupled with the efforts of some companies to achieve SDG 8 by 2030. Although this study is intended to guide the policies of countries in terms of labor issues, it is very enlightening and indicative insofar as it represents a step towards the psychosocial protection of workers.

2.3.3. Conflict-Related Causes of Absenteeism in the Hotel Sector

In view of the above, it seems clear that there are situations of conflict within organizations that can bring about illnesses caused by situations that lead to sick leave for those who suffer from them. This sick leave is one of the causes of absenteeism analyzed here.

For authors such as Durán (2010), absenteeism is defined as not attending the workplace during the established workday, either for justified (legal) or unjustified reasons, as well as not going to work for one or more days without being on vacation.

At the beginning of the studies on this aspect, some authors considered that all these assumptions other than the perceived lack of ability to perform the task and lack of motivation as a cause of absenteeism should be treated differently from absenteeism and consider the other causes as a mere response to a situation of job dissatisfaction (Steers and Rhodes 1978). For Addae and Johns (2012), the controversy is that all behaviors that involve job abandonment are usually analyzed. This is so because it is what causes harm to several affected parties, not only to the employees but also to the organizations and the different administrations, and therefore, no distinction is usually made.

Therefore, defining absenteeism is a task in which there is no unanimity, since it is a question of encompassing in a single concept something that has a great variety of behaviors and multiple causes. Curiously, this is the case with conflict.

Therefore, how individuals manage conflict in their work environment can lead to sick leave as a result of the psychosocial risks that the person may suffer (which would fall into what some researchers consider absenteeism). But it can also cause other types of situations involving absence from work or delays in entering or leaving work early because the worker does not want to go to work due to the conflictive situation he/she is experiencing at work.

In the scope of this research, the most accepted definition would be the one that understands absenteeism as "the non-presence of the worker and the lack of foresight of such absence by the employer" (Villaplana 2015), which is understood in Spain that it is due to occasional absences, excluding IT (temporary disabilities), vacations, and leaves.

Authors such as Evans and Walters (2002) state that TI is a situation caused by a determined behavior influenced by multiple factors, mainly psychosocial (Rael et al. 1995), that has to do with both the ability to go to work and the motivation to do so.

The statements of the president of the Spanish Confederation of Business Organizations (CEOE) in the prologue of Villaplana's (2015) are striking, in which he states that:

"Absenteeism is one of the most important problems in the world of labor relations in recent years. The employer suddenly notices that the workplaces are empty, with the consequent problems that this situation entails, without knowing the causes of these absences, and without being able to remedy them."

And the situation is indeed worrying due to the high cost to organizations, as discussed earlier in this study, since it is estimated that 23% of productivity loss in companies is due to absenteeism (Callen et al. 2013).

From a psychosocial point of view, absenteeism, or absenteeism as it is also known, is a phenomenon that highlights an underlying reality, that is, it reveals existing dysfunctions that may affect the workforce, hence the importance of its analysis with respect to conflict, since on many occasions this is also latent.

This leads us to think once again that there is another reality, and that is, in general, an adequate system for managing labor relations, conflict, and its resolution, which would contribute, to a large extent, to reducing the worrying rates of absenteeism in Spain, is still not being implemented in organizations. This occurs regardless of the existence of other

factors that cause this situation, but it is not dealt with because they are not the subject of this research.

Hence the study described below seeks to identify critical variables in the perception of the management of work relationships and conflict in organizations belonging to the hotel sector and helps determine whether the worker operates in a healthy environment and with quality work as intended in SDG 8.

## 3. Results

The research aims to validate which attributes have the greatest influence on the overall perception of the state of relationships and conflict management. In this case, the hypothesis to be validated is that the overall perception of relationships, and conflict management can be measured as a set of 57 explanatory variables or attributes, k, with attributes having different levels of impact on the overall perception.

**H$_0$.** *There are variables with a significant influence on the overall perception of labor relations and conflict for each variable i of the model where i $\in$ (0; m].*

**H$_{0,i}$.** *Variable i, related to the management of labor relations and conflict, influences the overall perception and is a critical variable.*

In order to identify the critical variables, the premises from which to start are as follows:

1. There are variables that have a greater influence on the overall perception expressed by the sampled population in the surveys conducted.
2. The most important attributes are those that contribute most to explaining the overall perception, i.e., variations in the perceptions given to these variables will generate oscillations in the overall perception scores. These attributes will be called critical variables.
3. There may be variables that have no influence or a very low influence on the overall perception. In other words, possible oscillations in the ratings given to these attributes will not generate a significant change in the overall perception. These variables will be referred to as non-significant or irrelevant variables.

In order to establish the critical variables and to contrast the hypotheses proposed previously, multiple linear regressions have been selected as the analysis technique. This technique allows us to quantify the relationship between a variable known as the dependent variable and a set of variables known as independent or explanatory variables. The dependent variable, in this case, will be the overall perception of each of the people interviewed on labor relations and conflict management in the hotel sector. The 57 attributes divided into the survey blocks have been used as independent variables. Linear regression has two main uses; one is to study whether the values assigned to one variable can be used to try to predict the values of another. In this case, we will analyze whether there are conflict-related variables that explain the overall perception. This will make it possible to determine and contrast the hypotheses, identify the critical variables, and determine how each of the independent variables positively or negatively influences the dependent variable and by how much. When applied to this research, it will allow quantifying the relative importance of each of the critical variables and will help in the development of actions to improve global perception in the organizations of the sector. The starting premise will be the following: the employed personnel give a score, not their global perception, which should be related to those attributes or variables critical for that person. The main purpose of a multiple linear regression model is to obtain an equation in which the sum of all the values given to each of the variables in each record or person employed, multiplied by a coefficient, yields a result that is as close as possible to the value of the overall perception. The model to be developed will have the following elements (Table 1):

**Table 1.** Multiple regression model.

| | Multiple Linear Regression Model—Notation |
|---|---|
| $y$ | Target variable. It is the variable whose result we are going to try to predict. In the case of the research, it will be the global perception. |
| $x_i$ | The explanatory variables are those variables that can be related to the object variable a priori. In the case of the research, they will be the 57 attributes that make up the management of labor relations and conflict. |
| $\hat{y}$ | Value of the variable is already estimated with the regression model. The model will have a notation similar to this. $$\hat{y} = \beta_0 + \beta_1 x_1 + \beta_2 x_2 + \ldots + \beta_m x_m + \varepsilon$$ where $m$ is the set of key attributes and $m \in (0; 57]$ |
| $\beta_0$ | Ordered in origin of the dependent variable, it is a constant value that does not vary in the case that the explanatory variables do. |
| $\beta_i$ | Slope. The sign indicates the direction of change in the object variable when there are changes in the associated independent variable. The magnitude shows the number of units the object variable changes for each unit of change in the independent variable. |
| $\varepsilon = \hat{y} - y$ | Error or residual. Difference between the value estimated using the model and the actual value collected in the investigation. The model seeks to minimize it. |

Source: Own elaboration (2023).

In order to perform a multiple linear regression model, two previous steps are required:

1. The response rate of the parameters to be introduced in the model must be contrasted. Its construction requires the participation of only those employees who have evaluated the dependent variable and at least one of the explanatory variables. In the present case, there are 92 employees in the hotel sector, who form the sample and meet the conditions. As in previous analyses, Monte Carlo simulations have been used to provide robustness to the results.

2. It is necessary to contrast the correlation between the participating parameters. Highly correlated attributes can generate distorting effects if the correlation between the different parameters is high. It is possible to perform a factor analysis that groups these attributes into factors that are independent of each other in order to neutralize the distorting effects of high correlation.

Correlation is a measure that indicates the strength and direction between two variables in the questionnaire. There is an acceptance of what is considered a high correlation, which is when the absolute value of the indicator exceeds 0.7 (Pérez 2013). In the correlation matrix, it is observed that only in six cases the partial correlation is higher than the reference value, and for this reason, a factor analysis will be discarded (Appendix B).

The last step, prior to performing the calculations, consists of selecting the procedure for the selection of the variables that are going to participate in the model. The stepwise algorithm has been selected, where the variables are introduced according to their importance and influence on the dependent variable in a progressive way, and those variables that lose importance when introducing new ones are removed. The reason is its high level of efficiency compared to other options; those variables of the model that do not significantly explain the changes in the dependent variable will be progressively eliminated (Pérez 2013). As a reference value to identify the significant variables, those with a p-value lower than 0.05 will be selected (Pérez 2013). The results obtained and an interpretation after the analysis performed are presented below. The resulting model expressed in the form of an equation is as follows: Table 2 (Set of variables included in the model) and Table 3 (Model Summary):

$$\hat{y} = -1.399 + 0.23x_1 + 0.13x_2 + 0.20x_3 + 0.14x_4 + 0.13x_5 + 0.25x_6 + 0.17x_7 - 0.16x_8 + 0.20x_9 + 0.12x_{10} + \varepsilon$$

- $x_1$ = Fair treatment irrespective of age
- $x_2$ = Recommendation to family/friends as a place of employment
- $x_3$ = The service is rated by the customer as excellent.
- $x_4$ = Contradictory orders from superiors on a regular basis.
- $x_5$ = The superiors represent the values of the company.
- $x_6$ = People are willing to go the extra mile to perform their job.
- $x_7$ = Processes established for day-to-day tasks.
- $x_8$ = Significant responsibilities for employees.
- $x_9$ = All employees are eligible for special recognition.
- $x_{10}$ = Superiors are unapproachable and difficult to talk to.

**Table 2.** Set of variables included in the model.

| Attribute | Coef No Std. | Error | Std Coef. | T | Sig. | Tol. | IVF |
|---|---|---|---|---|---|---|---|
| Constant | −1.399 | 0.344 | | −4.067 | 0 | | |
| We are treated fairly irrespective of our age | 0.232 | 0.065 | 0.249 | 3.569 | 0.001 | 0.545 | 1.835 |
| I would recommend family and friends to work here | 0.128 | 0.057 | 0.154 | 2.246 | 0.027 | 0.568 | 1.762 |
| Our customers would rate the service we deliver as "excellent" | 0.195 | 0.062 | 0.207 | 3.171 | 0.002 | 0.624 | 1.604 |
| It is common to receive contradictory orders from different managers | 0.139 | 0.045 | 0.179 | 3.127 | 0.002 | 0.815 | 1.227 |
| Superiors represent the values pursued by the company | 0.131 | 0.061 | 0.144 | 2.161 | 0.034 | 0.599 | 1.669 |
| People here are willing to go the extra mile to perform their job | 0.254 | 0.056 | 0.259 | 4.52 | 0 | 0.807 | 1.239 |
| Processes are in place for the performance of day-to-day tasks | 0.173 | 0.06 | 0.173 | 2.878 | 0.005 | 0.733 | 1.365 |
| People here are given a lot of responsibility | −0.163 | 0.057 | −0.173 | −2.873 | 0.005 | 0.737 | 1.358 |
| Here, we all have the opportunity to receive special recognition | 0.203 | 0.057 | 0.208 | 3.536 | 0.001 | 0.771 | 1.297 |
| Superiors are unapproachable and difficult to talk to | 0.116 | 0.048 | 0.142 | 2.449 | 0.016 | 0.795 | 1.258 |

Dependent variable: spontaneous global perception. Source: Own elaboration (2023).

**Table 3.** Model Summary.

| R | R Square | Adjusted R-Squared | Standard Error of the Estimate | Durbin-Watson |
|---|---|---|---|---|
| 0.883 | 0.779 | 0.753 | 0.511 | 2.063 |

Source: Own elaboration (2023).

The final model consists of 10 variables plus a constant. Each one has a significance level below the reference level, i.e., 0.05. In addition to indicating that they belong to the model, this value identifies them as being critical of the state of labor relations and conflict management and allows us to test the defined hypotheses, both the main one and those of a secondary nature. The global hypothesis (H0) has been validated: There are variables related to labor relations and conflict management that are more relevant in explaining changes in global perception.

- $H_{0.2}$: Accessibility by superiors significantly influences overall perception.
- $H_{0.4}$: Assignment of responsibilities to the people who collaborate in the organization significantly influences overall perception.
- $H_{0.6}$: Contradictory orders from superiors significantly influence overall perception.
- $H_{0.9}$: Establishment of processes for daily tasks significantly influences overall perception.

○ $H_{0.12}$: Representation of company values by superiors significantly influences overall perception.
○ $H_{0.31}$: Opportunity for all to receive special recognition significantly influences overall perception.
○ $H_{0.35}$: Fair treatment, regardless of age, significantly influences overall perception.
○ $H_{0.43}$: Willingness to go the extra mile to get the job done on the part of the people hired significantly influences overall perception.
○ $H_{0.45}$: Recommending the organization to family and friends as a place to work significantly influences the overall perception.
○ $H_{0.46}$: Customers' rating of the service being excellent has a significant influence on the overall perception.

The sign of the values in the coefficients of the variables indicates a positive contribution, i.e., an increase in the values or scores of these variables will increase global perception. In addition, this value indicates the extent to which global perception increases for each unit that is able to increase satisfaction with the independent variable. The quality of the model or model fit is determined using a given R-squared coefficient, which indicates the extent to which the variations in the critical variables explain the changes in the dependent variable. In this case, the R-squared coefficient is 77.9%, i.e., the model explains 77.9% of the overall variability in the perception of labor relations and conflict management, or, in other words, the remaining 22.1% is explained using variables that are not included in the model.

## 4. Materials and Methods

### 4.1. Sample

The target population of the study consists of people who work or have worked in the hotel sector during the last 5 years, amounting to an annual average of 209,017 in 2022 (Hosteltur 2023). For statistical purposes, this is considered an infinite population (Pérez 2009). The fact of establishing a period of 5 years in terms of their experience in the hotel sector is due to the fact that this is the time considered necessary in this research for a person employed in the sector to have sufficient knowledge of how a hotel works from the inside and what are the most common working practices. The subjects were selected based on simple random probability sampling without replacement, so the sample was made up of all the people who responded randomly to the questionnaire sent out. The use of this tool assumes that all units have the same probability of being selected as members of the sample (Pérez 2009). The number of completed surveys was 96, which yields a sampling error of 8%, below the 10% criterion established by researchers such as (López-Roldán and Fachelli 2015).

### 4.2. Data Collection

In order to respond to the set of objectives, quantitative research was chosen. For this purpose, the survey technique was used as a fundamental element of data collection, highlighting the different parts that make up the object of study (coping with conflict, work climate, internal communication, and engagement) and the characteristics of the sample in order to be able to adapt the research process to the circumstances. A questionnaire of sixty-nine questions has been prepared, the first part of which is aimed at finding out the different ways of coping with conflicts that employees working in a hotel experience. The instrument chosen for this study is a questionnaire based mainly on the Rahim Organizational Conflict Inventory (ROCI-II) by Rahim (1983), and important contributions have been made by adding another eight blocks of questions since the purpose was not only to investigate the style of coping with conflicts but also to analyze other variables that will be described in the following section and that contribute to a deeper analysis of the state of labor relations as a cause of conflict in the organizational environment, especially in the hotel sector to which it is addressed. The Great Place To Work (GPTW) questionnaire has also been taken as a basis, whose model, applied to organizations around the world, is very useful for

understanding the work climate and has been the subject of study by researchers such as Pérez Uribe (2012).

*4.3. Variables Analyzed*

A total of 58 variables were analyzed, distributed in 9 blocks that measure:

1. Overall perception of labor relations and conflict management.
2. Internal communication and work, in terms of work distribution and job-related requirements, entitled "MY SUPERVISORS AND MY JOB."
3. Internal communication and the alignment of the values of the person hired in the organization with those of the organization, entitled "MY COMPANY AND ITS VALUES."
4. The degree of commitment and recognition of the company towards the people who collaborate in it and the coordination, entitled "INVOLVEMENT AND RECOGNITION."
5. The work environment and its organization, entitled "ABOUT MY WORKPLACE AND FUNCTIONING."
6. The perception of fairness that the person working in the organization has with respect to treatment, salary, and professional development by the organization, entitled "HOW I FEEL THE TREATMENT TOWARDS ME."
7. The feeling of pride in belonging to the organization and the implication shown by the employee towards it, entitled "SENSE OF IDENTITY."
8. The relationship of the employee with the rest of the people who make up the organization and whether the work is carried out in a collaborative manner, entitled "CAMARADERIE."
9. The perception of the people employed in a hotel with respect to the company and their role within the company entitled, "PERCEPTION OF THE COMPANY".

In some of the blocks, it was decided to include questions of a negative nature, for if all the questions were formulated in the positive. There could be a bias of acquiescence in the answers that would vitiate the result. Each of the questions can be answered with one of the 5 established values since the Likert scale has been used in which 1 = Strongly Disagree, 2 = Disagree, 3 = Neutral, 4 = Agree, and 5 = Strongly Agree.

To facilitate the analysis of the data and the interpretation of the results in those variables that have a negative character in the field of labor relations, and in order to avoid acquiescence bias, a modification has been made to the scale, converting the high levels of agreement (fives) into disagreement (ones). Similarly, medium levels of disagreement (four) have been converted to agreement (two). The variables included in the analysis are included in Appendix A.

## 5. Discussion

The work presented here brings together theories related to conflict analysis systems, such as those proposed De Bono (1985), with the particularities of the working environment of the hotel sector, as well as establishing the relationship between labor relations and conflict management and SDG 8, which has not been addressed by the academy until now.

It is important to highlight the evolution of the study of conflict until it becomes unanimous in the theory that, if approached from a positive point of view (Yeung et al. 2015), it will positively affect both its management and the relationships in the organization's environment (Yang et al. 2019). It also reiterated the idea that conflicts arise from the moment there is human interaction, so companies in the hotel sector are no exception (Mohammad et al. 2018). This paper aims to contribute value to these companies, as well as to the academy in terms of the study of the management of labor relations and conflict insofar as it shows the perception that employees in the sector have of them and what variables hotel organizations can focus on to improve in this regard.

Through the study of the management of labor relations and conflict in the hotel sector, it is inferred that when employees are in a tense and conflictive work environment, productivity decreases, and this affects the quality of service (Ruizalba Robledo et al. 2015).

However, recent studies regarding the intentionality of abandonment or the manifestation of conflicts argue that this does not have a direct relationship in terms of the quality of service provided in hotels (Belias et al. 2022).

What is unanimous is the impact of poor labor relations and poor conflict management in relation to absenteeism and the cost it causes to the hotel organization and, therefore, to its economy. This is a phenomenon that can have a negative impact on the optimization of a hotel's human resources (Mukwevho et al. 2020), resources that are as important or more important than others, such as the need to train the emotional intelligence of employees and that it should not be undervalued (Webne-Behrman 1998; Barrientos-Báez 2018; Díaz Cuevas et al. 2021; Barrientos-Báez et al. 2022).

Likewise, if these situations occur, SDG 8, regarding the need for decent work for all, would be ignored. This objective, marked by the United Nations, should not only be analyzed from a wage perspective or the perspective of occupational risk prevention in the physical aspect but, as noted throughout this study, from the point of view of psychosocial risks and what they cause. Authors, such as (Barrientos-Báez et al. 2022), highlight the difficulty of reconciling the tools for the achievement of decent work for all with the economic growth encompassed in the same SDG since, for companies, this point tends to take precedence to the detriment of human capital. Hence, this research aims to shed light on the matter, identifying certain areas in which hotel companies can focus their efforts on improving the management of labor relations and conflict in order to, as the literature argues, generate much-needed competitive advantages in a sector that is so hard hit by the uncertainty caused by the recent COVID-19 pandemic or the recently unleashed wars.

## 6. Conclusions

This study has analyzed an entire sector without focusing on a specific conflict. By focusing on those variables that are not correctly managed or simply not managed in hotels, it is possible to extract those that are most relevant in terms of the possible generation of conflict and deterioration of labor relations.

The methodology employed has integrated two distinct stages: firstly, and after the pertinent literature review, qualitative research has been carried out by means of interviews with experts in the sector. The combination of both exercises has allowed us to develop exhaustive knowledge of the hotel sector, understand the management of labor relations and their impact on conflict, and validate the adequacy of the variables and indicators proposed in the objectives of this document.

In this work, a linear regression model has been developed to identify those variables that are critical in the perception of labor relations and conflict management. These variables account for less than 20% of the total model (10 variables included compared to the original 57 considered). Among them, it should be noted that there are also different levels of importance, i.e., changes in the level of agreement with these variables produce an impact of different magnitudes on the overall perception. As a result, it should be considered that there are 47 variables that are not critical or can be considered irrelevant, i.e., changes in the perception of employees on these variables have little impact on the overall perception of labor relations and conflict management.

The critical variables also explain 75% of the total changes in global perception. This indicator gives validity to the mathematical model and will allow hotel companies to direct their efforts and actions toward the variables with the greatest impact. An interesting line of research would be to try to relate the perception of these actions to the turnover rates of people employed in the hotel sector. Additionally, it is necessary to consider that there are still 25% of the changes in global perception that are explained using variables that are not contemplated in the model. Given this fact, it seems necessary to go deeper, through qualitative research, into the identification of other variables and items whose management could have a significant impact on global perception. It should be noted that the number of critical variables is specific to the hotel sector and may vary significantly in the application of this tool to other sectors or organizations.

It would be practical to go to a company and analyze it and work on the results obtained because each organization has its own resolution systems. It may even be detected that it does not have them. Only by first analyzing their systems and behavior will it be possible to see what works, the failures or shortcomings, and it will be possible to train their personnel, reinforcing the strengths and improving the failures. In the future, we intend to use quantitative techniques, such as a questionnaire, to a specific chain and complement it with qualitative techniques consisting of a series of in-depth interviews already focused on the conflict itself, according to the results of the analysis of the previous surveys. This will be conducted in order to, in future research, focus and complete the study to refine a questionnaire and focus group solely aimed at the conflict in the organization, adapted to the hotel chain in which the intervention is to be carried out. This study and its results are relevant for the Academy and society as a whole because an entire sector has been analyzed without focusing on a specific conflict. By focusing on those variables that are not managed correctly or are simply not managed in hotels, it is possible to extract those that are most relevant in terms of the possible generation of conflicts and deterioration of labor relations. This is relevant research for scientific knowledge because it highlights tourism, heritage, and the SDGs. The relevant results of this study will allow hotel companies to direct their efforts and actions toward variables with greater impact. Analyzing hotel organizations, their systems, and their behaviors has helped them to know what failures or deficiencies they have and, in this way, to train their professionals, reinforcing strengths and improving weaknesses.

**Author Contributions:** Conceptualization, M.d.C.P.-M.; methodology, J.A.V.-P., M.d.C.P.-M.; software, J.A.V.-P.; validation, J.A.V.-P.; formal analysis, J.A.V.-P.; investigation, J.A.V.-P., M.d.C.P.-M. and A.B.-B.; data curation, J.A.V.-P.; writing—original draft preparation, J.A.V.-P., M.d.C.P.-M. and A.B.-B.; writing—review and editing, M.d.C.P.-M., J.A.V.-P. and A.B.-B.; visualization, J.A.V.-P., M.d.C.P.-M. and A.B.-B.; supervision, M.d.C.P.-M., J.A.V.-P. and A.B.-B. All authors have read and agreed to the published version of the manuscript.

**Funding:** This research received no external funding.

**Data Availability Statement:** The data presented in this study are available on request from the corresponding author.

**Acknowledgments:** This article is part of the framework of a Concilium project (931.791) of the Complutense University of Madrid, "Validation of communication models, neurocommunication, business, social networks and gender".

**Conflicts of Interest:** The authors declare no conflict of interest.

## Appendix A. Questionnaire (Investigation of the Management of Labor Relations and Conflict)

Information from superiors on important issues and changes.

○ Accessibility by superiors.
○ Assignment of functions and coordination of teams by superiors.
○ Assignment of responsibilities to the people who collaborate in the organization.
○ Fulfillment of promises by superiors.
○ Contradictory orders from superiors.
○ Clarity in the assignment of objectives for the area or department.
○ Actual performance of job duties.
○ Establishment of processes for daily tasks.
○ Coordination between departments in case of possible events.
○ Communication of the organization's values and mission.
○ Representation of the company's values by superiors.
○ Alignment between personal and business values.
○ Alignment between personal values and those of the rest of the team.
○ Training provided for professional growth.

- ○ Allocation of resources and equipment to perform the assigned work.
- ○ Recognition by superiors of extra work and effort.
- ○ Disincentivization by superiors of ideas and suggestions.
- ○ Consideration of ideas and suggestions from superiors.
- ○ Sincere response from superiors to ideas and suggestions.
- ○ Involvement of the teams in the decisions affecting them made by their superiors.
- ○ Generation of problems derived from the way each member of the organization works.
- ○ Existence of different points of view in equal situations.
- ○ The site is physically safe to work in.
- ○ The place is emotionally healthy to work in.
- ○ Help is provided to balance personal and professional life.
- ○ Existence of shift changes at the last moment depending on the workload.
- ○ Double shifts depending on workload.
- ○ Interest in the member of the organization as a person and not only as an employee.
- ○ Unfair payment for work performed.
- ○ Opportunity for all to receive special recognition.
- ○ Reception of good treatment regardless of the position in the company.
- ○ Promotions awarded to the most deserving employees.
- ○ Absence of "politicking" or "cheap shots" in the achievement of objectives.
- ○ Fair treatment regardless of age.
- ○ Fair treatment regardless of race, ethnicity or religion.
- ○ Fair treatment regardless of physical condition.
- ○ Fair treatment regardless of sexual orientation.
- ○ Possibility to complain, to be heard and to receive adequate treatment in the event of being treated unfairly.
- ○ Job stability in the workplace.
- ○ Endorsement from family and friends of the company's excellence as a place to work.
- ○ Quick adaptation to changes for the success of the company by its members.
- ○ Willingness to go the extra mile to get the job done on the part of the people hired.
- ○ I feel ashamed when I say that I work for this company.
- ○ Recommending the organization to family and friends as a place to work.
- ○ Customers rate the service provided as excellent.
- ○ Concern of the people working in the organization for each other.
- ○ Sense of welcome when employees join the organization.
- ○ Possibility of counting on the collaboration of the rest of the members when someone needs it.
- ○ Sense of real teamwork in all situations.
- ○ Existence of competition between people working as partners.
- ○ Perception of the company's agility and efficiency in implementing change.
- ○ Making the most of the skills, knowledge and experience of the members of the organization.
- ○ Transmission of dissatisfaction with the work of collaborators when it occurs.
- ○ Transmission of possible areas for improvement by superiors.
- ○ Career support within the company from superiors.
- ○ Perception of the possibilities for career advancement within the organization.

## Appendix B

**Table A1.** Correlation Matrix.

| | | | | | | | | | | | | | | | | | | | | | | | | | | | | | | | | | | | | | | | | | | | | | | | | | | | | | | | | |
|---|---|---|---|---|---|---|---|---|---|---|---|---|---|---|---|---|---|---|---|---|---|---|---|---|---|---|---|---|---|---|---|---|---|---|---|---|---|---|---|---|---|---|---|---|---|---|---|---|---|---|---|---|---|---|---|---|
| 1.0 | 0.3 | 0.4 | 0.4 | 0.4 | 0.1 | 0.5 | 0.4 | 0.4 | 0.4 | 0.4 | 0.4 | 0.1 | 0.0 | 0.4 | 0.6 | 0.4 | 0.1 | 0.4 | 0.4 | 0.4 | −0.2 | −0.3 | 0.4 | 0.2 | 0.1 | −0.1 | 0.0 | 0.4 | 0.1 | 0.2 | 0.4 | 0.2 | 0.1 | 0.5 | 0.3 | 0.3 | 0.4 | 0.3 | 0.1 | 0.3 | 0.1 | 0.1 | 0.4 | 0.4 | 0.4 | 0.5 | 0.4 | 0.6 | 0.3 | 0.1 | 0.2 | 0.3 | 0.2 | 0.4 | 0.3 | 0.3 |
| 0.3 | 1.0 | 0.3 | 0.2 | 0.2 | 0.2 | 0.3 | 0.2 | 0.2 | 0.2 | 0.4 | 0.4 | 0.1 | 0.2 | 0.3 | 0.2 | 0.2 | 0.5 | 0.4 | 0.3 | 0.3 | 0.1 | 0.1 | 0.5 | 0.2 | 0.2 | 0.3 | 0.2 | 0.3 | 0.4 | 0.1 | 0.3 | 0.1 | 0.2 | 0.3 | 0.2 | 0.3 | 0.3 | 0.2 | 0.1 | 0.2 | 0.0 | 0.2 | 0.4 | 0.2 | 0.2 | 0.2 | 0.2 | 0.2 | 0.4 | 0.3 | 0.0 | 0.2 | 0.1 | 0.2 | 0.3 | 0.2 |
| 0.4 | 0.3 | 1.0 | 0.3 | 0.6 | 0.2 | 0.3 | 0.2 | 0.2 | 0.3 | 0.2 | 0.5 | 0.1 | 0.1 | 0.5 | 0.4 | 0.4 | 0.2 | 0.4 | 0.5 | 0.4 | 0.0 | 0.1 | 0.3 | 0.3 | 0.2 | 0.1 | 0.1 | 0.4 | 0.3 | 0.3 | 0.3 | 0.3 | 0.1 | 0.4 | 0.3 | 0.4 | 0.3 | 0.2 | 0.1 | 0.4 | 0.1 | 0.0 | 0.3 | 0.4 | 0.2 | 0.1 | 0.3 | 0.4 | 0.2 | 0.1 | 0.3 | 0.2 | 0.0 | 0.1 | 0.3 | 0.3 |
| 0.4 | 0.2 | 0.3 | 1.0 | 0.4 | 0.1 | 0.3 | 0.1 | 0.3 | 0.2 | 0.3 | 0.4 | 0.1 | 0.2 | 0.2 | 0.3 | 0.3 | 0.1 | 0.4 | 0.4 | 0.4 | −0.2 | −0.1 | 0.1 | 0.1 | 0.2 | 0.0 | 0.2 | 0.3 | 0.0 | 0.1 | 0.2 | 0.3 | 0.0 | 0.3 | 0.2 | 0.3 | 0.3 | 0.3 | 0.1 | 0.1 | 0.2 | 0.4 | 0.3 | 0.2 | 0.2 | 0.3 | 0.3 | 0.4 | 0.3 | 0.0 | 0.0 | 0.4 | 0.2 | 0.2 | 0.3 | 0.2 |
| 0.4 | 0.2 | 0.6 | 0.4 | 1.0 | 0.4 | 0.4 | 0.4 | 0.3 | 0.3 | 0.2 | 0.5 | 0.1 | 0.1 | 0.6 | 0.5 | 0.6 | 0.2 | 0.5 | 0.6 | 0.4 | 0.0 | 0.1 | 0.2 | 0.4 | 0.3 | 0.1 | 0.2 | 0.4 | 0.4 | 0.3 | 0.4 | 0.4 | 0.2 | 0.4 | 0.3 | 0.4 | 0.3 | 0.4 | 0.3 | 0.5 | 0.2 | 0.0 | 0.3 | 0.4 | 0.3 | 0.2 | 0.4 | 0.4 | 0.2 | 0.1 | 0.3 | 0.2 | 0.1 | 0.2 | 0.5 | 0.4 |
| 0.1 | 0.2 | 0.2 | 0.1 | 0.4 | 1.0 | 0.2 | 0.3 | 0.1 | 0.1 | 0.0 | 0.1 | 0.1 | −0.1 | 0.3 | 0.3 | 0.4 | 0.3 | 0.3 | 0.3 | 0.2 | 0.3 | 0.2 | 0.2 | 0.3 | 0.3 | 0.3 | 0.1 | 0.3 | 0.4 | 0.2 | 0.3 | 0.3 | 0.1 | 0.1 | 0.2 | 0.2 | 0.1 | 0.3 | 0.1 | 0.3 | 0.1 | 0.0 | 0.1 | 0.4 | 0.1 | 0.0 | 0.1 | 0.1 | 0.2 | 0.2 | 0.1 | 0.1 | −0.2 | 0.1 | 0.3 | 0.2 |
| 0.5 | 0.3 | 0.3 | 0.3 | 0.4 | 0.2 | 1.0 | 0.6 | 0.5 | 0.3 | 0.6 | 0.4 | 0.1 | 0.1 | 0.3 | 0.5 | 0.3 | 0.3 | 0.4 | 0.4 | 0.4 | 0.0 | −0.3 | 0.5 | 0.3 | 0.2 | 0.1 | 0.2 | 0.4 | 0.1 | 0.2 | 0.4 | 0.2 | 0.0 | 0.5 | 0.4 | 0.5 | 0.6 | 0.4 | 0.3 | 0.4 | 0.3 | 0.2 | 0.6 | 0.4 | 0.3 | 0.4 | 0.5 | 0.5 | 0.2 | 0.1 | 0.1 | 0.1 | 0.1 | 0.3 | 0.3 | 0.3 |
| 0.4 | 0.2 | 0.2 | 0.1 | 0.4 | 0.3 | 0.6 | 1.0 | 0.3 | 0.5 | 0.4 | 0.4 | 0.1 | 0.0 | 0.4 | 0.5 | 0.3 | 0.2 | 0.4 | 0.4 | 0.3 | 0.2 | 0.0 | 0.3 | 0.3 | 0.3 | 0.1 | 0.3 | 0.3 | 0.2 | 0.3 | 0.5 | 0.2 | 0.1 | 0.4 | 0.2 | 0.3 | 0.3 | 0.3 | 0.4 | 0.5 | 0.2 | 0.1 | 0.4 | 0.5 | 0.3 | 0.3 | 0.4 | 0.4 | 0.2 | 0.2 | 0.3 | 0.2 | −0.1 | 0.2 | 0.4 | 0.4 |
| 0.4 | 0.2 | 0.2 | 0.3 | 0.3 | 0.1 | 0.5 | 0.3 | 1.0 | 0.4 | 0.3 | 0.3 | −0.1 | −0.1 | 0.2 | 0.4 | 0.1 | 0.1 | 0.3 | 0.3 | 0.3 | −0.2 | −0.3 | 0.4 | 0.3 | 0.3 | 0.0 | 0.1 | 0.2 | 0.0 | 0.1 | 0.2 | 0.1 | 0.1 | 0.5 | 0.3 | 0.5 | 0.5 | 0.3 | 0.1 | 0.2 | 0.3 | 0.2 | 0.4 | 0.1 | 0.4 | 0.3 | 0.3 | 0.4 | 0.2 | 0.2 | 0.1 | 0.1 | 0.1 | 0.3 | 0.2 | 0.1 |
| 0.4 | 0.2 | 0.3 | 0.2 | 0.3 | 0.1 | 0.3 | 0.5 | 0.4 | 1.0 | 0.2 | 0.3 | −0.1 | 0.0 | 0.3 | 0.2 | 0.2 | 0.2 | 0.3 | 0.4 | 0.2 | 0.1 | 0.0 | 0.3 | 0.3 | 0.2 | 0.0 | 0.1 | 0.1 | 0.3 | 0.2 | 0.2 | 0.3 | 0.2 | 0.4 | 0.2 | 0.2 | 0.3 | 0.2 | 0.3 | 0.3 | 0.3 | 0.3 | 0.4 | 0.3 | 0.3 | 0.4 | 0.3 | 0.4 | 0.3 | 0.2 | 0.3 | 0.1 | 0.1 | 0.1 | 0.2 | 0.2 |
| 0.4 | 0.4 | 0.2 | 0.3 | 0.2 | 0.0 | 0.6 | 0.4 | 0.3 | 0.2 | 1.0 | 0.5 | 0.2 | 0.2 | 0.4 | 0.3 | 0.2 | 0.2 | 0.4 | 0.3 | 0.3 | 0.0 | −0.2 | 0.5 | 0.2 | 0.2 | 0.1 | 0.3 | 0.3 | 0.1 | 0.1 | 0.4 | 0.0 | 0.0 | 0.4 | 0.3 | 0.3 | 0.4 | 0.3 | 0.3 | 0.4 | 0.2 | 0.2 | 0.5 | 0.3 | 0.4 | 0.4 | 0.4 | 0.3 | 0.2 | 0.0 | 0.2 | 0.3 | 0.1 | 0.4 | 0.2 | 0.3 |
| 0.4 | 0.4 | 0.5 | 0.4 | 0.5 | 0.1 | 0.4 | 0.4 | 0.3 | 0.3 | 0.5 | 1.0 | 0.0 | 0.1 | 0.6 | 0.4 | 0.4 | 0.2 | 0.5 | 0.5 | 0.4 | −0.1 | 0.0 | 0.3 | 0.2 | 0.3 | 0.0 | 0.2 | 0.4 | 0.3 | 0.3 | 0.4 | 0.2 | 0.1 | 0.4 | 0.2 | 0.3 | 0.3 | 0.3 | 0.2 | 0.5 | 0.2 | 0.1 | 0.4 | 0.4 | 0.4 | 0.3 | 0.4 | 0.4 | 0.3 | 0.2 | 0.4 | 0.4 | 0.1 | 0.4 | 0.4 | 0.3 |
| 0.1 | 0.1 | 0.1 | 0.1 | 0.1 | 0.1 | 0.1 | 0.1 | −0.1 | −0.1 | 0.2 | 0.0 | 1.0 | 0.3 | 0.1 | 0.2 | 0.2 | 0.2 | 0.3 | 0.1 | 0.1 | 0.2 | 0.1 | 0.1 | 0.2 | 0.1 | 0.0 | 0.0 | 0.3 | 0.1 | 0.2 | 0.2 | 0.1 | 0.2 | 0.2 | 0.0 | 0.2 | 0.2 | 0.0 | 0.0 | 0.2 | 0.2 | 0.2 | 0.2 | 0.2 | 0.2 | 0.2 | 0.2 | 0.3 | 0.2 | 0.0 | 0.0 | 0.1 | 0.2 | 0.1 | 0.1 | 0.2 |
| 0.0 | 0.2 | 0.1 | 0.2 | 0.1 | −0.1 | 0.1 | 0.0 | −0.1 | 0.0 | 0.2 | 0.1 | 0.3 | 1.0 | 0.0 | 0.0 | 0.1 | 0.3 | 0.1 | 0.2 | 0.1 | 0.1 | 0.0 | 0.1 | 0.1 | 0.1 | 0.0 | 0.0 | 0.2 | 0.1 | 0.2 | 0.1 | 0.1 | 0.0 | 0.1 | 0.0 | 0.2 | 0.2 | 0.0 | 0.1 | 0.1 | 0.0 | 0.2 | 0.3 | 0.1 | 0.1 | 0.1 | 0.1 | 0.1 | 0.2 | 0.2 | 0.0 | 0.0 | 0.0 | 0.1 | 0.1 | 0.0 |
| 0.4 | 0.3 | 0.5 | 0.2 | 0.6 | 0.3 | 0.3 | 0.4 | 0.2 | 0.3 | 0.4 | 0.6 | 0.1 | 0.0 | 1.0 | 0.5 | 0.5 | 0.3 | 0.5 | 0.4 | 0.4 | 0.0 | 0.1 | 0.2 | 0.3 | 0.4 | 0.1 | 0.3 | 0.4 | 0.3 | 0.3 | 0.5 | 0.3 | 0.2 | 0.4 | 0.3 | 0.3 | 0.3 | 0.4 | 0.3 | 0.5 | 0.1 | 0.0 | 0.2 | 0.4 | 0.3 | 0.3 | 0.4 | 0.4 | 0.2 | 0.3 | 0.4 | 0.3 | 0.1 | 0.4 | 0.4 | 0.5 |
| 0.6 | 0.2 | 0.4 | 0.3 | 0.5 | 0.3 | 0.5 | 0.5 | 0.4 | 0.2 | 0.3 | 0.4 | 0.2 | 0.0 | 0.5 | 1.0 | 0.6 | 0.3 | 0.6 | 0.5 | 0.5 | 0.0 | −0.1 | 0.4 | 0.4 | 0.4 | 0.1 | 0.2 | 0.6 | 0.3 | 0.4 | 0.6 | 0.2 | 0.0 | 0.6 | 0.5 | 0.5 | 0.5 | 0.4 | 0.2 | 0.5 | 0.2 | 0.1 | 0.4 | 0.5 | 0.4 | 0.4 | 0.5 | 0.5 | 0.3 | 0.1 | 0.2 | 0.3 | 0.0 | 0.5 | 0.4 | 0.4 |
| 0.4 | 0.2 | 0.4 | 0.3 | 0.6 | 0.4 | 0.3 | 0.3 | 0.1 | 0.2 | 0.2 | 0.4 | 0.2 | 0.1 | 0.5 | 0.6 | 1.0 | 0.3 | 0.6 | 0.5 | 0.5 | 0.1 | 0.0 | 0.3 | 0.4 | 0.3 | 0.2 | 0.1 | 0.5 | 0.4 | 0.5 | 0.5 | 0.3 | 0.3 | 0.4 | 0.3 | 0.3 | 0.2 | 0.4 | 0.2 | 0.5 | 0.1 | 0.1 | 0.2 | 0.4 | 0.2 | 0.2 | 0.2 | 0.2 | 0.3 | 0.0 | 0.2 | 0.2 | 0.0 | 0.3 | 0.5 | 0.4 |
| 0.1 | 0.5 | 0.2 | 0.1 | 0.2 | 0.3 | 0.3 | 0.2 | 0.1 | 0.2 | 0.2 | 0.2 | 0.2 | 0.3 | 0.3 | 0.3 | 0.3 | 1.0 | 0.3 | 0.3 | 0.2 | 0.2 | 0.2 | 0.3 | 0.2 | 0.2 | 0.3 | 0.1 | 0.2 | 0.4 | 0.1 | 0.3 | 0.2 | 0.2 | 0.2 | 0.2 | 0.3 | 0.3 | 0.3 | 0.2 | 0.2 | 0.0 | 0.1 | 0.3 | 0.2 | 0.1 | 0.2 | 0.2 | 0.2 | 0.2 | 0.4 | 0.0 | 0.1 | 0.1 | 0.2 | 0.2 | 0.2 |
| 0.4 | 0.4 | 0.4 | 0.4 | 0.5 | 0.3 | 0.4 | 0.4 | 0.3 | 0.3 | 0.4 | 0.5 | 0.3 | 0.1 | 0.5 | 0.6 | 0.6 | 0.3 | 1.0 | 0.6 | 0.6 | 0.0 | −0.1 | 0.4 | 0.3 | 0.5 | 0.1 | 0.1 | 0.6 | 0.3 | 0.5 | 0.6 | 0.3 | 0.3 | 0.5 | 0.3 | 0.5 | 0.5 | 0.3 | 0.3 | 0.5 | 0.2 | 0.1 | 0.3 | 0.5 | 0.3 | 0.4 | 0.5 | 0.4 | 0.4 | 0.1 | 0.2 | 0.4 | 0.1 | 0.5 | 0.6 | 0.5 |
| 0.4 | 0.3 | 0.5 | 0.4 | 0.6 | 0.3 | 0.4 | 0.4 | 0.3 | 0.4 | 0.3 | 0.5 | 0.1 | 0.2 | 0.4 | 0.5 | 0.5 | 0.3 | 0.6 | 1.0 | 0.6 | 0.1 | 0.0 | 0.4 | 0.4 | 0.4 | 0.2 | 0.1 | 0.5 | 0.4 | 0.6 | 0.5 | 0.4 | 0.2 | 0.5 | 0.3 | 0.5 | 0.4 | 0.4 | 0.4 | 0.6 | 0.3 | 0.2 | 0.4 | 0.5 | 0.4 | 0.3 | 0.5 | 0.4 | 0.3 | 0.2 | 0.3 | 0.3 | 0.2 | 0.3 | 0.4 | 0.3 |
| 0.4 | 0.3 | 0.4 | 0.4 | 0.4 | 0.2 | 0.4 | 0.3 | 0.3 | 0.2 | 0.3 | 0.4 | 0.1 | 0.1 | 0.4 | 0.5 | 0.5 | 0.2 | 0.6 | 0.6 | 1.0 | 0.0 | 0.0 | 0.3 | 0.3 | 0.3 | 0.1 | 0.0 | 0.5 | 0.1 | 0.5 | 0.4 | 0.3 | 0.2 | 0.3 | 0.2 | 0.3 | 0.3 | 0.3 | 0.2 | 0.3 | 0.1 | 0.0 | 0.1 | 0.3 | 0.1 | 0.1 | 0.3 | 0.2 | 0.2 | 0.0 | 0.2 | 0.2 | 0.2 | 0.3 | 0.4 | 0.3 |
| −0.2 | 0.1 | 0.0 | −0.2 | 0.0 | 0.3 | 0.0 | 0.2 | −0.2 | 0.1 | 0.0 | −0.1 | 0.2 | 0.1 | 0.0 | 0.0 | 0.1 | 0.2 | 0.0 | 0.1 | 0.0 | 1.0 | 0.5 | 0.1 | 0.2 | 0.2 | 0.3 | 0.0 | 0.1 | 0.2 | 0.1 | 0.0 | 0.1 | 0.3 | 0.0 | 0.1 | 0.0 | −0.1 | 0.1 | 0.1 | 0.2 | 0.3 | 0.1 | 0.0 | 0.1 | 0.0 | −0.1 | −0.1 | 0.0 | 0.2 | 0.3 | 0.2 | 0.1 | −0.2 | 0.0 | 0.2 | 0.2 |
| −0.3 | 0.1 | 0.1 | −0.1 | 0.1 | 0.2 | −0.3 | 0.0 | −0.3 | 0.0 | −0.2 | 0.0 | 0.1 | 0.0 | 0.1 | −0.1 | 0.0 | 0.2 | −0.1 | 0.0 | 0.0 | 0.5 | 1.0 | −0.2 | 0.1 | 0.1 | 0.2 | 0.1 | −0.1 | 0.2 | −0.1 | 0.0 | −0.1 | 0.2 | −0.3 | −0.1 | −0.3 | −0.3 | −0.1 | −0.1 | −0.1 | 0.1 | 0.0 | −0.2 | −0.1 | −0.2 | −0.2 | −0.3 | −0.2 | 0.0 | 0.2 | 0.1 | 0.1 | −0.1 | −0.2 | 0.1 | 0.0 |
| 0.4 | 0.5 | 0.3 | 0.1 | 0.2 | 0.2 | 0.5 | 0.3 | 0.4 | 0.3 | 0.5 | 0.3 | 0.1 | 0.1 | 0.2 | 0.4 | 0.3 | 0.3 | 0.4 | 0.4 | 0.3 | 0.1 | −0.2 | 1.0 | 0.4 | 0.3 | 0.2 | 0.1 | 0.2 | 0.3 | 0.2 | 0.3 | 0.2 | 0.1 | 0.5 | 0.4 | 0.6 | 0.5 | 0.4 | 0.2 | 0.4 | 0.2 | 0.2 | 0.6 | 0.3 | 0.4 | 0.4 | 0.4 | 0.5 | 0.3 | 0.1 | 0.0 | 0.0 | 0.0 | 0.2 | 0.2 | 0.3 |
| 0.2 | 0.2 | 0.3 | 0.1 | 0.4 | 0.3 | 0.3 | 0.3 | 0.3 | 0.3 | 0.2 | 0.2 | 0.2 | 0.1 | 0.3 | 0.4 | 0.4 | 0.2 | 0.3 | 0.4 | 0.3 | 0.2 | 0.1 | 0.4 | 1.0 | 0.6 | 0.1 | 0.1 | 0.5 | 0.3 | 0.5 | 0.3 | 0.4 | 0.4 | 0.4 | 0.4 | 0.4 | 0.3 | 0.4 | 0.5 | 0.6 | 0.5 | 0.1 | 0.3 | 0.6 | 0.4 | 0.3 | 0.2 | 0.4 | 0.4 | 0.2 | 0.3 | 0.2 | 0.0 | 0.3 | 0.5 | 0.5 |
| 0.1 | 0.2 | 0.2 | 0.2 | 0.3 | 0.3 | 0.2 | 0.3 | 0.3 | 0.2 | 0.2 | 0.3 | 0.1 | 0.1 | 0.4 | 0.4 | 0.3 | 0.2 | 0.5 | 0.4 | 0.3 | 0.2 | 0.1 | 0.3 | 0.6 | 1.0 | 0.3 | 0.2 | 0.5 | 0.4 | 0.5 | 0.4 | 0.2 | 0.4 | 0.3 | 0.4 | 0.5 | 0.4 | 0.5 | 0.4 | 0.5 | 0.3 | 0.1 | 0.2 | 0.6 | 0.2 | 0.3 | 0.4 | 0.3 | 0.4 | 0.3 | 0.3 | 0.3 | 0.0 | 0.4 | 0.4 | 0.4 |
| −0.1 | 0.3 | 0.1 | 0.0 | 0.1 | 0.3 | 0.1 | 0.1 | 0.0 | 0.0 | 0.1 | 0.0 | 0.0 | 0.0 | 0.1 | 0.1 | 0.2 | 0.3 | 0.1 | 0.2 | 0.1 | 0.3 | 0.2 | 0.2 | 0.1 | 0.3 | 1.0 | 0.4 | 0.1 | 0.3 | 0.1 | 0.2 | 0.1 | 0.2 | 0.0 | 0.1 | 0.1 | 0.0 | 0.2 | 0.1 | 0.1 | 0.1 | 0.0 | 0.1 | 0.0 | 0.0 | 0.0 | −0.1 | 0.0 | 0.0 | 0.4 | −0.1 | 0.0 | −0.2 | −0.1 | 0.1 | 0.0 |
| 0.0 | 0.2 | 0.1 | 0.2 | 0.2 | 0.1 | 0.2 | 0.3 | 0.1 | 0.1 | 0.3 | 0.2 | 0.0 | 0.0 | 0.3 | 0.2 | 0.1 | 0.1 | 0.1 | 0.1 | 0.0 | 0.0 | 0.1 | 0.1 | 0.1 | 0.2 | 0.4 | 1.0 | 0.2 | 0.2 | 0.0 | 0.4 | 0.1 | 0.0 | 0.0 | 0.1 | 0.0 | 0.0 | 0.2 | 0.2 | 0.2 | 0.2 | 0.2 | 0.2 | 0.2 | 0.0 | 0.2 | 0.1 | 0.1 | 0.1 | 0.3 | 0.1 | 0.2 | −0.1 | 0.0 | 0.1 | 0.1 |
| 0.4 | 0.3 | 0.4 | 0.3 | 0.4 | 0.3 | 0.4 | 0.3 | 0.2 | 0.1 | 0.3 | 0.4 | 0.3 | 0.2 | 0.4 | 0.6 | 0.5 | 0.2 | 0.6 | 0.5 | 0.5 | 0.1 | −0.1 | 0.2 | 0.5 | 0.5 | 0.1 | 0.2 | 1.0 | 0.2 | 0.6 | 0.5 | 0.3 | 0.3 | 0.4 | 0.3 | 0.4 | 0.4 | 0.3 | 0.3 | 0.6 | 0.1 | 0.1 | 0.2 | 0.5 | 0.2 | 0.3 | 0.4 | 0.4 | 0.4 | 0.1 | 0.2 | 0.1 | 0.0 | 0.4 | 0.6 | 0.5 |
| 0.1 | 0.4 | 0.3 | 0.0 | 0.4 | 0.4 | 0.1 | 0.2 | 0.0 | 0.3 | 0.1 | 0.3 | 0.1 | 0.1 | 0.3 | 0.3 | 0.4 | 0.4 | 0.3 | 0.4 | 0.1 | 0.2 | 0.2 | 0.3 | 0.3 | 0.4 | 0.3 | 0.2 | 0.2 | 1.0 | 0.2 | 0.3 | 0.3 | 0.2 | 0.2 | 0.3 | 0.2 | 0.2 | 0.2 | 0.2 | 0.3 | 0.0 | 0.1 | 0.2 | 0.4 | 0.2 | 0.1 | 0.2 | 0.2 | 0.3 | 0.2 | 0.1 | 0.1 | 0.0 | 0.1 | 0.2 | 0.1 |
| 0.2 | 0.1 | 0.3 | 0.1 | 0.3 | 0.2 | 0.2 | 0.3 | 0.1 | 0.2 | 0.1 | 0.3 | 0.2 | 0.2 | 0.3 | 0.4 | 0.5 | 0.1 | 0.5 | 0.6 | 0.5 | 0.1 | −0.1 | 0.2 | 0.5 | 0.5 | 0.1 | 0.0 | 0.6 | 0.2 | 1.0 | 0.4 | 0.3 | 0.4 | 0.4 | 0.2 | 0.3 | 0.2 | 0.3 | 0.4 | 0.6 | 0.2 | 0.0 | 0.2 | 0.4 | 0.2 | 0.2 | 0.3 | 0.2 | 0.3 | 0.2 | 0.2 | 0.1 | 0.1 | 0.3 | 0.4 | 0.4 |
| 0.4 | 0.3 | 0.3 | 0.2 | 0.4 | 0.3 | 0.4 | 0.5 | 0.2 | 0.2 | 0.4 | 0.4 | 0.2 | 0.1 | 0.5 | 0.6 | 0.5 | 0.3 | 0.6 | 0.5 | 0.4 | 0.0 | 0.0 | 0.3 | 0.3 | 0.4 | 0.2 | 0.4 | 0.5 | 0.3 | 0.4 | 1.0 | 0.2 | 0.2 | 0.4 | 0.4 | 0.4 | 0.4 | 0.4 | 0.4 | 0.5 | 0.1 | 0.1 | 0.2 | 0.5 | 0.2 | 0.3 | 0.5 | 0.3 | 0.3 | 0.2 | 0.3 | 0.3 | 0.1 | 0.4 | 0.5 | 0.4 |
| 0.2 | 0.1 | 0.3 | 0.3 | 0.4 | 0.3 | 0.2 | 0.2 | 0.1 | 0.3 | 0.0 | 0.2 | 0.1 | 0.1 | 0.3 | 0.2 | 0.3 | 0.2 | 0.3 | 0.4 | 0.3 | 0.1 | −0.1 | 0.2 | 0.4 | 0.2 | 0.1 | 0.1 | 0.3 | 0.3 | 0.3 | 0.2 | 1.0 | 0.3 | 0.3 | 0.1 | 0.3 | 0.1 | 0.3 | 0.4 | 0.4 | 0.1 | 0.3 | 0.1 | 0.3 | 0.1 | 0.2 | 0.2 | 0.3 | 0.3 | 0.1 | 0.1 | 0.2 | −0.1 | 0.2 | 0.4 | 0.4 |
| 0.1 | 0.2 | 0.1 | 0.0 | 0.2 | 0.1 | 0.0 | 0.1 | 0.1 | 0.2 | 0.0 | 0.1 | 0.2 | 0.0 | 0.2 | 0.0 | 0.3 | 0.2 | 0.3 | 0.2 | 0.2 | 0.3 | 0.2 | 0.1 | 0.4 | 0.4 | 0.2 | 0.0 | 0.3 | 0.2 | 0.4 | 0.2 | 0.3 | 1.0 | 0.2 | 0.1 | 0.2 | 0.1 | 0.1 | 0.2 | 0.2 | 0.3 | 0.3 | 0.1 | 0.2 | 0.1 | 0.2 | 0.1 | 0.2 | 0.4 | 0.3 | 0.1 | 0.1 | 0.1 | 0.3 | 0.3 | 0.3 |
| 0.5 | 0.3 | 0.4 | 0.3 | 0.4 | 0.1 | 0.5 | 0.4 | 0.5 | 0.4 | 0.4 | 0.4 | 0.2 | 0.1 | 0.4 | 0.6 | 0.4 | 0.2 | 0.5 | 0.5 | 0.3 | 0.0 | −0.3 | 0.5 | 0.4 | 0.3 | 0.0 | 0.0 | 0.4 | 0.2 | 0.4 | 0.4 | 0.3 | 0.2 | 1.0 | 0.7 | 0.8 | 0.7 | 0.5 | 0.4 | 0.5 | 0.3 | 0.3 | 0.6 | 0.5 | 0.5 | 0.5 | 0.6 | 0.6 | 0.4 | 0.1 | 0.1 | 0.2 | 0.3 | 0.5 | 0.3 | 0.3 |
| 0.3 | 0.2 | 0.3 | 0.2 | 0.3 | 0.2 | 0.4 | 0.2 | 0.3 | 0.2 | 0.3 | 0.2 | 0.0 | 0.0 | 0.3 | 0.5 | 0.3 | 0.2 | 0.3 | 0.3 | 0.2 | 0.1 | −0.1 | 0.4 | 0.4 | 0.4 | 0.1 | 0.1 | 0.3 | 0.3 | 0.2 | 0.4 | 0.1 | 0.1 | 0.7 | 1.0 | 0.6 | 0.6 | 0.6 | 0.4 | 0.4 | 0.1 | 0.2 | 0.4 | 0.4 | 0.4 | 0.3 | 0.4 | 0.4 | 0.2 | 0.0 | 0.2 | 0.1 | 0.2 | 0.4 | 0.2 | 0.3 |
| 0.3 | 0.3 | 0.4 | 0.3 | 0.4 | 0.2 | 0.5 | 0.3 | 0.5 | 0.2 | 0.3 | 0.3 | 0.2 | 0.2 | 0.3 | 0.5 | 0.3 | 0.3 | 0.5 | 0.5 | 0.3 | 0.0 | −0.3 | 0.6 | 0.4 | 0.5 | 0.1 | 0.0 | 0.4 | 0.2 | 0.3 | 0.4 | 0.3 | 0.2 | 0.8 | 0.6 | 1.0 | 0.7 | 0.5 | 0.4 | 0.4 | 0.3 | 0.2 | 0.5 | 0.4 | 0.4 | 0.4 | 0.6 | 0.6 | 0.4 | 0.2 | 0.1 | 0.1 | 0.1 | 0.4 | 0.4 | 0.4 |
| 0.4 | 0.3 | 0.3 | 0.3 | 0.3 | 0.1 | 0.6 | 0.3 | 0.5 | 0.3 | 0.4 | 0.3 | 0.2 | 0.2 | 0.3 | 0.5 | 0.2 | 0.3 | 0.5 | 0.4 | 0.3 | −0.1 | −0.3 | 0.5 | 0.3 | 0.4 | 0.0 | 0.0 | 0.4 | 0.2 | 0.2 | 0.4 | 0.1 | 0.1 | 0.7 | 0.6 | 0.7 | 1.0 | 0.5 | 0.4 | 0.4 | 0.2 | 0.2 | 0.5 | 0.3 | 0.4 | 0.5 | 0.7 | 0.6 | 0.4 | 0.2 | 0.0 | 0.1 | 0.2 | 0.4 | 0.2 | 0.3 |
| 0.3 | 0.2 | 0.2 | 0.3 | 0.4 | 0.3 | 0.4 | 0.3 | 0.3 | 0.2 | 0.3 | 0.3 | 0.0 | 0.0 | 0.4 | 0.4 | 0.4 | 0.3 | 0.3 | 0.4 | 0.3 | 0.1 | −0.1 | 0.4 | 0.4 | 0.5 | 0.2 | 0.2 | 0.3 | 0.2 | 0.3 | 0.4 | 0.3 | 0.1 | 0.5 | 0.6 | 0.5 | 0.5 | 1.0 | 0.5 | 0.5 | 0.3 | 0.2 | 0.4 | 0.6 | 0.4 | 0.4 | 0.5 | 0.4 | 0.2 | 0.2 | 0.3 | 0.3 | 0.1 | 0.3 | 0.2 | 0.3 |
| 0.1 | 0.1 | 0.1 | 0.1 | 0.3 | 0.1 | 0.3 | 0.4 | 0.1 | 0.3 | 0.3 | 0.2 | 0.0 | 0.1 | 0.3 | 0.2 | 0.2 | 0.2 | 0.3 | 0.4 | 0.2 | 0.1 | −0.1 | 0.2 | 0.5 | 0.4 | 0.1 | 0.2 | 0.3 | 0.2 | 0.4 | 0.4 | 0.4 | 0.2 | 0.4 | 0.4 | 0.4 | 0.4 | 0.5 | 1.0 | 0.4 | 0.3 | 0.1 | 0.3 | 0.4 | 0.3 | 0.4 | 0.4 | 0.4 | 0.3 | 0.3 | 0.1 | 0.1 | 0.2 | 0.2 | 0.4 | 0.4 |
| 0.3 | 0.2 | 0.4 | 0.1 | 0.5 | 0.3 | 0.4 | 0.5 | 0.2 | 0.3 | 0.4 | 0.5 | 0.2 | 0.1 | 0.5 | 0.5 | 0.5 | 0.2 | 0.5 | 0.6 | 0.3 | 0.2 | −0.1 | 0.4 | 0.6 | 0.5 | 0.1 | 0.2 | 0.6 | 0.3 | 0.6 | 0.5 | 0.4 | 0.2 | 0.5 | 0.4 | 0.4 | 0.4 | 0.5 | 0.4 | 1.0 | 0.3 | 0.1 | 0.4 | 0.8 | 0.4 | 0.3 | 0.5 | 0.3 | 0.3 | 0.1 | 0.5 | 0.3 | 0.0 | 0.4 | 0.5 | 0.5 |
| 0.1 | 0.0 | 0.1 | 0.2 | 0.2 | 0.1 | 0.3 | 0.2 | 0.3 | 0.3 | 0.2 | 0.2 | 0.2 | 0.0 | 0.1 | 0.2 | 0.1 | 0.0 | 0.2 | 0.3 | 0.1 | 0.3 | 0.1 | 0.2 | 0.5 | 0.3 | 0.1 | 0.2 | 0.1 | 0.0 | 0.2 | 0.1 | 0.1 | 0.3 | 0.3 | 0.1 | 0.3 | 0.2 | 0.3 | 0.3 | 0.3 | 1.0 | 0.4 | 0.2 | 0.2 | 0.4 | 0.3 | 0.2 | 0.4 | 0.4 | 0.2 | 0.2 | 0.3 | 0.1 | 0.2 | 0.2 | 0.2 |
| 0.1 | 0.2 | 0.0 | 0.4 | 0.0 | 0.0 | 0.2 | 0.1 | 0.2 | 0.3 | 0.2 | 0.1 | 0.2 | 0.2 | 0.0 | 0.1 | 0.1 | 0.1 | 0.1 | 0.2 | 0.0 | 0.1 | 0.0 | 0.2 | 0.1 | 0.1 | 0.0 | 0.2 | 0.1 | 0.1 | 0.0 | 0.1 | 0.3 | 0.3 | 0.3 | 0.2 | 0.2 | 0.2 | 0.2 | 0.1 | 0.1 | 0.4 | 1.0 | 0.3 | 0.2 | 0.2 | 0.4 | 0.1 | 0.3 | 0.4 | 0.1 | 0.0 | 0.2 | 0.1 | 0.1 | 0.0 | 0.0 |
| 0.4 | 0.4 | 0.3 | 0.3 | 0.3 | 0.1 | 0.6 | 0.4 | 0.4 | 0.4 | 0.5 | 0.4 | 0.2 | 0.3 | 0.2 | 0.4 | 0.2 | 0.3 | 0.3 | 0.4 | 0.1 | 0.0 | −0.2 | 0.6 | 0.3 | 0.2 | 0.1 | 0.2 | 0.2 | 0.2 | 0.2 | 0.2 | 0.1 | 0.1 | 0.6 | 0.4 | 0.5 | 0.5 | 0.4 | 0.3 | 0.4 | 0.2 | 0.3 | 1.0 | 0.4 | 0.6 | 0.5 | 0.5 | 0.6 | 0.3 | 0.3 | 0.1 | 0.2 | 0.1 | 0.3 | 0.1 | 0.1 |
| 0.4 | 0.2 | 0.4 | 0.2 | 0.4 | 0.4 | 0.4 | 0.5 | 0.1 | 0.3 | 0.3 | 0.4 | 0.2 | 0.1 | 0.4 | 0.5 | 0.4 | 0.2 | 0.5 | 0.5 | 0.3 | 0.1 | −0.1 | 0.3 | 0.6 | 0.6 | 0.0 | 0.2 | 0.5 | 0.4 | 0.4 | 0.5 | 0.3 | 0.2 | 0.5 | 0.4 | 0.4 | 0.3 | 0.6 | 0.4 | 0.8 | 0.2 | 0.2 | 0.4 | 1.0 | 0.4 | 0.4 | 0.5 | 0.4 | 0.4 | 0.1 | 0.5 | 0.3 | 0.0 | 0.3 | 0.4 | 0.5 |
| 0.4 | 0.2 | 0.2 | 0.2 | 0.3 | 0.1 | 0.3 | 0.3 | 0.4 | 0.3 | 0.4 | 0.4 | 0.2 | 0.1 | 0.3 | 0.4 | 0.2 | 0.1 | 0.3 | 0.4 | 0.1 | 0.0 | −0.2 | 0.4 | 0.4 | 0.2 | 0.0 | 0.0 | 0.2 | 0.2 | 0.2 | 0.2 | 0.1 | 0.1 | 0.5 | 0.4 | 0.4 | 0.4 | 0.4 | 0.3 | 0.4 | 0.4 | 0.2 | 0.6 | 0.4 | 1.0 | 0.5 | 0.4 | 0.5 | 0.4 | 0.1 | 0.3 | 0.2 | 0.2 | 0.2 | 0.1 | 0.1 |
| 0.5 | 0.2 | 0.1 | 0.3 | 0.2 | 0.0 | 0.4 | 0.3 | 0.3 | 0.4 | 0.4 | 0.3 | 0.2 | 0.1 | 0.3 | 0.4 | 0.2 | 0.2 | 0.4 | 0.3 | 0.1 | −0.1 | −0.2 | 0.4 | 0.3 | 0.3 | 0.0 | 0.2 | 0.3 | 0.1 | 0.2 | 0.3 | 0.2 | 0.2 | 0.5 | 0.3 | 0.4 | 0.5 | 0.4 | 0.4 | 0.3 | 0.3 | 0.4 | 0.5 | 0.4 | 0.5 | 1.0 | 0.6 | 0.7 | 0.6 | 0.2 | 0.2 | 0.3 | 0.2 | 0.3 | 0.3 | 0.4 |
| 0.4 | 0.2 | 0.3 | 0.3 | 0.4 | 0.1 | 0.5 | 0.4 | 0.3 | 0.3 | 0.4 | 0.4 | 0.2 | 0.1 | 0.4 | 0.5 | 0.2 | 0.2 | 0.5 | 0.5 | 0.3 | −0.1 | −0.3 | 0.4 | 0.2 | 0.4 | −0.1 | 0.1 | 0.4 | 0.2 | 0.3 | 0.5 | 0.2 | 0.1 | 0.6 | 0.4 | 0.6 | 0.7 | 0.5 | 0.4 | 0.5 | 0.2 | 0.1 | 0.5 | 0.5 | 0.4 | 0.6 | 1.0 | 0.6 | 0.4 | 0.3 | 0.2 | 0.2 | 0.1 | 0.4 | 0.3 | 0.4 |
| 0.6 | 0.2 | 0.4 | 0.4 | 0.4 | 0.1 | 0.5 | 0.4 | 0.4 | 0.4 | 0.3 | 0.4 | 0.3 | 0.1 | 0.4 | 0.5 | 0.2 | 0.2 | 0.4 | 0.4 | 0.2 | 0.0 | −0.2 | 0.5 | 0.4 | 0.3 | 0.0 | 0.1 | 0.4 | 0.2 | 0.2 | 0.3 | 0.3 | 0.2 | 0.6 | 0.4 | 0.6 | 0.6 | 0.4 | 0.4 | 0.3 | 0.4 | 0.3 | 0.6 | 0.4 | 0.5 | 0.7 | 0.6 | 1.0 | 0.6 | 0.2 | 0.1 | 0.3 | 0.1 | 0.3 | 0.3 | 0.4 |
| 0.3 | 0.4 | 0.2 | 0.3 | 0.2 | 0.2 | 0.2 | 0.2 | 0.2 | 0.3 | 0.2 | 0.3 | 0.2 | 0.2 | 0.2 | 0.3 | 0.3 | 0.2 | 0.4 | 0.3 | 0.2 | 0.2 | 0.0 | 0.3 | 0.4 | 0.4 | 0.0 | 0.1 | 0.4 | 0.3 | 0.3 | 0.3 | 0.3 | 0.4 | 0.4 | 0.2 | 0.4 | 0.4 | 0.2 | 0.3 | 0.3 | 0.4 | 0.4 | 0.3 | 0.4 | 0.4 | 0.6 | 0.4 | 0.6 | 1.0 | 0.2 | 0.1 | 0.3 | 0.0 | 0.2 | 0.3 | 0.3 |
| 0.1 | 0.3 | 0.1 | 0.0 | 0.1 | 0.2 | 0.1 | 0.2 | 0.2 | 0.2 | 0.0 | 0.2 | 0.0 | 0.2 | 0.3 | 0.1 | 0.0 | 0.4 | 0.1 | 0.2 | 0.0 | 0.3 | 0.2 | 0.1 | 0.2 | 0.3 | 0.4 | 0.3 | 0.1 | 0.2 | 0.2 | 0.2 | 0.1 | 0.3 | 0.1 | 0.0 | 0.2 | 0.2 | 0.2 | 0.3 | 0.1 | 0.2 | 0.1 | 0.3 | 0.1 | 0.1 | 0.2 | 0.3 | 0.2 | 0.2 | 1.0 | 0.0 | 0.1 | −0.1 | 0.2 | 0.1 | 0.0 |
| 0.2 | 0.0 | 0.3 | 0.0 | 0.3 | 0.1 | 0.1 | 0.3 | 0.1 | 0.3 | 0.2 | 0.4 | 0.0 | 0.0 | 0.4 | 0.2 | 0.2 | 0.0 | 0.2 | 0.3 | 0.2 | 0.2 | 0.1 | 0.0 | 0.3 | 0.3 | −0.1 | 0.1 | 0.2 | 0.1 | 0.2 | 0.3 | 0.1 | 0.1 | 0.1 | 0.2 | 0.1 | 0.0 | 0.3 | 0.1 | 0.5 | 0.2 | 0.0 | 0.1 | 0.5 | 0.3 | 0.2 | 0.2 | 0.1 | 0.1 | 0.0 | 1.0 | 0.3 | 0.0 | 0.3 | 0.2 | 0.2 |
| 0.3 | 0.2 | 0.2 | 0.4 | 0.2 | 0.1 | 0.1 | 0.2 | 0.1 | 0.1 | 0.3 | 0.4 | 0.1 | 0.0 | 0.3 | 0.3 | 0.2 | 0.1 | 0.4 | 0.3 | 0.2 | 0.1 | 0.1 | 0.0 | 0.2 | 0.3 | 0.0 | 0.2 | 0.1 | 0.1 | 0.1 | 0.3 | 0.2 | 0.1 | 0.2 | 0.1 | 0.1 | 0.1 | 0.3 | 0.1 | 0.3 | 0.3 | 0.2 | 0.2 | 0.3 | 0.2 | 0.3 | 0.2 | 0.3 | 0.3 | 0.1 | 0.3 | 1.0 | 0.2 | 0.3 | 0.3 | 0.3 |
| 0.2 | 0.1 | 0.0 | 0.2 | 0.1 | −0.2 | 0.1 | −0.1 | 0.1 | 0.1 | 0.1 | 0.1 | 0.2 | 0.0 | 0.1 | 0.0 | 0.0 | 0.1 | 0.1 | 0.2 | 0.2 | −0.2 | −0.1 | 0.0 | 0.0 | 0.0 | −0.2 | −0.1 | 0.0 | 0.0 | 0.1 | 0.1 | −0.1 | 0.1 | 0.3 | 0.2 | 0.1 | 0.2 | 0.1 | 0.2 | 0.0 | 0.1 | 0.1 | 0.1 | 0.0 | 0.2 | 0.2 | 0.1 | 0.1 | 0.0 | −0.1 | 0.0 | 0.2 | 1.0 | 0.4 | 0.0 | 0.0 |
| 0.4 | 0.2 | 0.1 | 0.2 | 0.2 | 0.1 | 0.3 | 0.2 | 0.3 | 0.1 | 0.4 | 0.4 | 0.1 | 0.1 | 0.4 | 0.5 | 0.3 | 0.2 | 0.5 | 0.3 | 0.3 | 0.0 | −0.2 | 0.2 | 0.3 | 0.4 | −0.1 | 0.0 | 0.4 | 0.1 | 0.3 | 0.4 | 0.2 | 0.3 | 0.5 | 0.4 | 0.4 | 0.4 | 0.3 | 0.2 | 0.4 | 0.2 | 0.1 | 0.3 | 0.3 | 0.2 | 0.3 | 0.4 | 0.3 | 0.2 | 0.2 | 0.3 | 0.3 | 0.4 | 1.0 | 0.4 | 0.3 |
| 0.3 | 0.3 | 0.3 | 0.3 | 0.5 | 0.3 | 0.3 | 0.4 | 0.2 | 0.2 | 0.2 | 0.4 | 0.1 | 0.1 | 0.4 | 0.4 | 0.5 | 0.2 | 0.6 | 0.4 | 0.4 | 0.2 | 0.1 | 0.2 | 0.5 | 0.4 | 0.1 | 0.1 | 0.6 | 0.2 | 0.4 | 0.5 | 0.4 | 0.3 | 0.3 | 0.2 | 0.4 | 0.2 | 0.2 | 0.4 | 0.5 | 0.2 | 0.0 | 0.1 | 0.4 | 0.1 | 0.3 | 0.3 | 0.3 | 0.3 | 0.1 | 0.2 | 0.3 | 0.0 | 0.4 | 1.0 | 0.7 |
| 0.3 | 0.2 | 0.3 | 0.2 | 0.4 | 0.2 | 0.3 | 0.4 | 0.1 | 0.2 | 0.3 | 0.3 | 0.2 | 0.0 | 0.5 | 0.4 | 0.4 | 0.2 | 0.5 | 0.3 | 0.3 | 0.2 | 0.0 | 0.3 | 0.5 | 0.4 | 0.0 | 0.1 | 0.5 | 0.1 | 0.4 | 0.4 | 0.4 | 0.3 | 0.3 | 0.3 | 0.4 | 0.3 | 0.3 | 0.4 | 0.5 | 0.2 | 0.0 | 0.1 | 0.5 | 0.1 | 0.4 | 0.4 | 0.4 | 0.3 | 0.0 | 0.2 | 0.3 | 0.0 | 0.3 | 0.7 | 1.0 |

Source: Own elaboration (2023).

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
