# Peer review of "Model for the Identification of Key Elements in the Management of Labor Relations and Conflict: Impact on the Internal Customer of Hotel Organizations and on Sustainable Development Goals 8"

_admsci, doi:10.3390/admsci13120252_

Round 1
Reviewer 1 Report
Comments and Suggestions for Authors
The objective of this article is to study conflict in the workplace within tourism organizations, specifically those dedicated to the hotel sector, to understand the variables that affect it and to try to identify the importance of each one of them in the general perception of conflict.
This paper contributes to identify in a broad way those variables that have an influence on conflict management by organizations and through an empirical study allows to identify among these, those that are critical in the perception of conflict and the management of labor relations by workers in the hotel sector, providing organizations with effective tools to avoid it or face it in a constructive way, and create positive environments.
The hypothesis of this paper arises from the theoretical context reviewed, from which it is clear that in organizations there are a number of factors that directly influence labor relations and conflict, and that, if they are well valued by the people working in the company, it will lead to a competitive advantage over others. In addition, how the people hired by the company perceive certain variables will provide an overall view of how they feel about their work, how much they are willing to give of themselves and the degree of attention that, in this case the hotel sector, is paid to human capital.
The methodology employed has integrated two distinct stages: firstly, and after the pertinent literature review, qualitative research has been carried out by means of interviews with experts in the sector. The combination of both exercises has allowed us to develop an exhaustive knowledge of the hotel sector, to understand the management of labor relations and their impact on conflict, as well as to validate the adequacy of the variables and indicators proposed in the objectives of this document. In this work, a linear regression model has been developed to identify those variables that are critical in the perception of labor relations and conflict management. Derived from this fact, it should be considered that there are variables that are not critical or can be considered as irrelevant, i.e., changes in the perception of employees on these variables have little impact on the overall perception of labor relations and conflict management.
The critical variables also allow explaining 75% of the total changes in the global perception, this indicator gives validity to the mathematical model and will allow hotel companies to direct their efforts and actions towards the variables with the greatest impact.
The article is new and interesting enough to justify publication.
The connection is made between the subject of the paper and that of the previous studies, making bibliographical references from the specialized literature.
A description of the current state of knowledge in the field takes place, in a clear, systematic, critical, coherent and concise way, compared to previous or recent achievements.
Considering the above, I recommend the authors to consider in the future articles the papers considered relevant, especially those in the main flow of publications.
The authors did not indicate how the results are in relation to previous expectations and research.
The authors do not explain how research is a step forward for scientific knowledge.
Author Response
Thank you very much for the comments and the exhaustive analysis that Reviewer 1 has carried out on our research. We have added in the Conclusions section the explanation of what was suggested by the reviewer. We hope it turned out as you requested in your review.
Reviewer 2 Report
Comments and Suggestions for Authors
This article deals with a remarkably interesting and outstanding topic, namely the research on conflict in the workplace within hotel companies to orient its efforts towards the fulfilment of the SDGs.
The article responds adequately to the common scientific structure, highlighting its methodological rigour based on an extensive questionnaire that allows for the identification of the variables that most influence conflict in labour relations.
The title “Model for the Identification of key elements in the management of labor relations and conflict: impact on the internal customer of hotel organizations and on SDG 8” accurately reflects the content and purpose of the paper. The abstract is concise and provides sufficient information. The keywords are adequate.
The introduction section presents a very good framework which locates well the work, followed by an extensive analysis of the theoretical framework, supported by a broad scientific literature. The article is notable for the large number of bibliographical references, most of which have been published in the last 5 years and are therefore highly topical.
The article correctly describes the methodology and research methods, which are established in total coherence with the conceptual definition previously explained, highlighting the relevance of the items to be assessed. As a consequence, the results are consistent and represent a contribution to a better knowledge of relationships and conflict management in hotel companies.
The article correctly describes the methodology and research methods, which are established in total coherence with the conceptual definition previously explained, highlighting the high number and the relevance of the items to be assessed. As a consequence, the results are consistent and represent an innovative contribution to a better knowledge of relationships and conflict management in hotel companies.
I would also like to point out that, beyond its scientific relevance, this study stands out for its social commitment, seeking to contribute to improving working conditions in one of the main economic sectors worldwide. The article thus seeks to transcend the exclusive sphere of academia, proposing applied research with a high potential for transfer to society, and whose methodology can be extrapolated to other contexts.
For all these reasons, the article should be accepted in its present form.
Author Response
Many thanks to reviewer 2 for his detailed comments. THANKS to your analysis you have motivated us to continue working in this same line of work.
Reviewer 3 Report
Comments and Suggestions for Authors
The purpose of the presented paper is to analyse conflicts within hotel companies, exploring their impact on various issues such as operations, employee-company relations, and overall satisfaction. This research caters to both academia and businesses seeking alignment with the Sustainable Development Goals (SDGs) outlined in the 2030 Agenda, specifically addressing SDG 8 - decent work for all.
The study identifies key areas where proactive measures should be implemented to mitigate conflicts among stakeholders, thereby fostering the well-being of company members. This, in turn, is anticipated to enhance competitiveness and yield greater economic benefits. The methodology employed is robust, organized into distinct phases including a literature review, expert consultations, and data analysis utilizing a linear regression model.
This original work holds significance for hotel companies dedicated to SDGs. Despite the topic having been explored in previous studies, this paper delves into novel inquiries related to contemporary global issues, previously untouched. It stands as a valuable reference for future research and, as emphasized by the authors, lays the groundwork for subsequent studies employing new techniques and methodologies, particularly focusing on individual company assessments and employee perceptions.
The method applied is coherent and well structured, based on a series of well-established phases. A suggestion is made to further explore the uniqueness of each company, acknowledging that this aspect is already identified as an area for future research within the text.
The tables (1, 2, 3, and B1) complement the arguments presented, serving as valuable tools for contrasting information. Additionally, the appendices contribute to the utility of the paper for future research endeavours by corroborating the information provided in the text.
The conclusions align seamlessly with the study's content, addressing the set objectives and paving the way for new avenues of research and future actions to delve deeper into the discussed aspects.
A potential improvement could involve incorporating relevant existing papers to enhance the empirical basis:
- Díaz Cuevas, M.P.; Becerra Fernández, D.; Villar Lama, A. Transición Ecológica y Emergencia Climática en las enseñanzas de Turismo. Cuadernos de Turismo, 2021, 48, 325 - 349. DOI: https://doi.org/10.6018/turismo.492791
- Mayer, B. The dynamics of conflict resolution. A practitioner´s guide. Jossey-Bass, United States of America, 2000.
- Webne-Behrman, H. The practice of facilitation. Managing group process and solving problems. United States of America, Quorum Books, 1998.
Author Response
Many thanks to reviewer 2 for his detailed comments. We have introduced two of the three references that he suggested.